# Memory is Reconstructed, Not Retrieved: Graph Memory for LLM Agents

**Shuo Ji**[1]  **Yibo Li**[1]  **Bryan Hooi**[1]

## Abstract

Despite recent progress, LLM agents still struggle with reasoning over long interaction histories. While current memory-augmented agents rely on a static "retrieve-then-reason" paradigm, this rigid pipeline design prevents them from dynamically adapting memory access to intermediate evidence discovered during inference. To bridge this gap, we propose MRAgent, a framework that combines an associative memory graph with an active reconstruction mechanism. We represent memory as a Cue–Tag–Content graph, where associative tags serve as semantic bridges connecting fine-grained cues to memory contents. Operating on this structure, our active reconstruction mechanism integrates LLM reasoning directly into memory access, allowing the agent to iteratively explore and prune retrieval paths based on accumulated evidence. This ensures that memory retrieval is dynamically adapted to the reasoning context while avoiding combinatorial explosion caused by unconstrained expansion. Experiments on the LoCoMo benchmark and LongMemEval benchmark demonstrate significant improvements over strong baselines (up to 23%), while substantially reducing token and runtime cost, highlighting the effectiveness of active and associative reconstruction for long-horizon memory reasoning. The code is available at: https://github.com/Ji-shuo/MRAgent.

## 1. Introduction

LLMs exhibit a "jagged" cognitive profile (Hendrycks et al., 2025), excelling at math and reasoning, but deficient in tasks requiring long-term memory, such as interactive assistance or decision-support systems acting over extended interactions (Gao et al., 2026). In such long-horizon tasks, LLMs

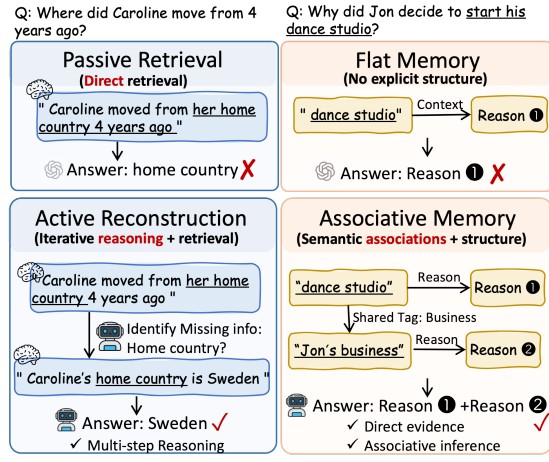

*Figure 1.* Comparison between passive retrieval and active memory reconstruction in MRAgent.

are fundamentally constrained by their limited context windows, which restrict their ability to retain interaction history over time (Hatalis et al., 2023).

To mitigate these limitations, prior work equips LLM agents with external memory systems. Early approaches adopt Retrieval-Augmented Generation (RAG) (Lewis et al., 2020), where memory access is realized via similarity-based retrieval over unstructured text or embedding stores. Subsequent work introduces more structured memory representations, including hierarchical stores (Fang et al., 2025; Kang et al., 2025) and knowledge graphs (KGs) (Rasmussen et al., 2025; Xu et al., 2025; Huang et al., 2025), which explicitly encode entities and relations to support more interpretable and relational retrieval. However, retrieval in these systems remains restricted to fixed top-$k$ selection or predefined subgraph traversal, failing to infer new retrieval cues or adapt to intermediate evidence. Under this formulation, existing memory systems operate as *passive retrieval policies*.

In contrast, cognitive neuroscience conceptualizes memory retrieval as an *active* and *associative* reconstruction process (Rugg & Renoult, 2025), rather than a passive read-out of stored content. Specifically, retrieval is initiated by contextual cues, which propagate through intermediate representations, progressively reconstructing coherent memory experiences (Frankland & Josselyn, 2019).

[1]National University of Singapore, Singapore. Correspondence to: Yibo Li <liyibo@u.nus.edu>.

*Proceedings of the 43rd International Conference on Machine Learning*, Seoul, South Korea. PMLR 306, 2026. Copyright 2026 by the author(s).

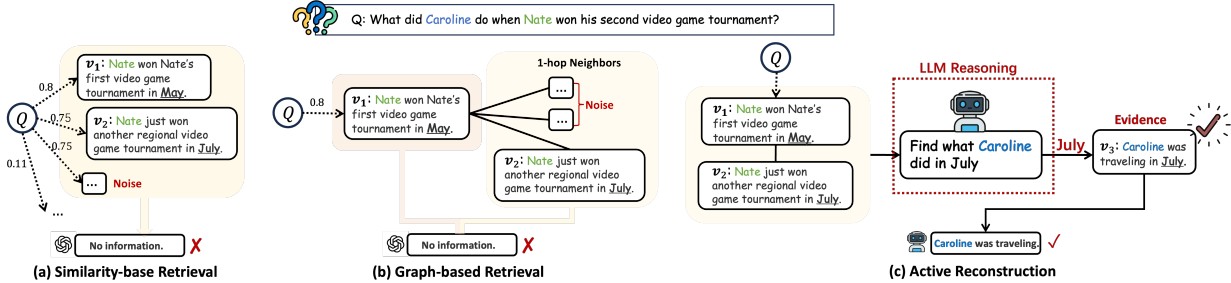

*Figure 2.* Illustrative example comparing passive retrieval and active reconstruction: passive retrieval only retrieves memory content related to Nate's video game tournaments based on the query, while active reconstruction infers a critical temporal cue ("July") through LLM reasoning and identifies Caroline's corresponding activity.

This perspective highlights two key challenges for LLM-based memory systems, as illustrated in Figure 1. **(Challenge 1: Active Reconstruction)** How can memory access be transformed from one-shot retrieval into an active multi-step reconstruction process that progressively reveals information across reasoning steps? **(Challenge 2: Associative Memory Structure)** How can memory be organized to capture semantic and structural dependencies, enabling guided exploration across associative items?

In this light, we propose the **M**emory **R**easoning Architecture for LLM **Agent**s (MRAgent), a framework that enables LLM agents to perform active and associative memory reconstruction by integrating LLM reasoning into memory access. MRAgent explores multiple candidate retrieval paths over a memory graph, pruning irrelevant branches based on intermediate evidence, and iteratively optimizing for the next step with the highest information gain. To enable controlled memory traversal, we design a Cue–Tag–Content memory graph, where tags encode associations between fine-grained cues and specific memory contents. By exposing these associations explicitly, tags allow the LLM to select retrieval paths before accessing detailed memory contents. Our contributions can be summarized as follows:

- We propose active memory reconstruction, a new paradigm that integrates memory access into the reasoning process, allowing the agent to dynamically adapt search strategy based on intermediate evidence.

- We introduce a Cue–Tag–Content memory graph in which associative tags mediate retrieval between cues and content, enabling the LLM to identify promising retrieval paths while pruning irrelevant branches.

- We provide a theoretical analysis proving that active retrieval policies are strictly more expressive than passive retrieval.

- Extensive experiments demonstrate that MRAgent outperforms strong baselines with significantly improved token and runtime efficiency.

## 2. Problem Setting: Active Memory Access

In this section, we formulate memory retrieval as a sequential decision process and analyze why the traditional passive retrieval paradigm fails on complex, multi-step queries.

### 2.1. Memory Retrieval Definition

We consider an external memory $\mathcal{M}$ consisting of memory units $\mathcal{V} = \{v_1, \ldots, v_N\}$. Given a query $x$, memory access proceeds by sequentially selecting memory units over $T$ steps. Let $S^{(t)} = \{v^{(1)}, \ldots, v^{(t)}\}$ denote the accumulated evidence after step $t$. We distinguish two classes of memory access policies.

**Passive retrieval.** A *passive* (stateless) retrieval policy $\pi_{\mathrm{p}}$ selects memory units solely based on the query:

$$\{v^{(1)}, \ldots, v^{(T)}\} = \pi_{\mathrm{p}}(x). \tag{1}$$

**Active reconstruction.** An *active* (stateful) policy $\pi_{\mathrm{a}}^{(t)}$ selects the next unit conditioned on the evolving evidence:

$$v^{(t)} = \pi_{\mathrm{a}}^{(t)}(x, S^{(t-1)}), \qquad S^{(t)} = S^{(t-1)} \cup \{v^{(t)}\}. \tag{2}$$

This formulation enables adaptive, multi-step interaction with memory.

### 2.2. Limits of Passive Retrieval

We categorize retrieval strategies of existing memory systems into two paradigms based on their memory organization and adaptation mechanisms, and compare them with the proposed active reconstruction paradigm in Figure 2.

**Similarity-based Retrieval.** Many existing memory systems adopt similarity-based retrieval, such as Memory-Bank (Zhong et al., 2024) and Mem0 (Chhikara et al., 2025), where retrieval is defined by a similarity scoring function:

$$\pi_{\mathrm{sim}}(x) = \mathrm{TopK}\left(\{\mathrm{sim}(x, v)\}_{v \in \mathcal{V}}, k\right). \tag{3}$$

In this paradigm, memory serves as a static context provider and the relevance function $\mathrm{sim}(x, v)$ is fixed by the query.

As shown in Figure 2(a), such methods retrieve many events related to "video game tournament" based on surface-level relevance, thereby introducing substantial noise while failing to find the correct evidence.

**Graph-based Memory Retrieval.** To exploit structural relations among memory units, methods such as A-Mem (Xu et al., 2025), Zep (Rasmussen et al., 2025) introduce graph-structured memory. These approaches extend the retrieval mechanism by combining similarity-based seeding with pre-defined $N$-hop neighbor expansion:

$$\mathcal{V}^{\text{sim}} = \text{TopK}\left(\{\text{sim}(x, v)\}_{v \in \mathcal{V}}, k\right),$$
$$\pi_{\text{graph}}(x) = \mathcal{V}^{\text{sim}} \cup \text{Neighbor}\left(\mathcal{V}^{\text{sim}}\right). \tag{4}$$

where $\text{Neighbor}(\cdot)$ returns the predefined $N$-hop neighbors of a node set in the memory graph. While this strategy alleviates certain multi-hop retrieval challenges, it requires relevant evidence to be connected through explicit graph links. Moreover, the use of fixed neighbor expansion often introduces substantial noise. Figure 2(b) shows that graph-based expansion retrieves additional but irrelevant neighbors, yet still fails to recover information about Caroline, which is not directly connected to "video game tournament" events in the memory graph.

**Active Memory Reconstruction.** By integrating LLM reasoning directly into memory access, active reconstruction enables the agent to infer new retrieval cues and adapt traversal strategies based on accumulated evidence. As illustrated in Figure 2(c), intermediate findings can be explicitly transformed into new retrieval constraints, allowing the agent to recover evidence that is unreachable under passive policies.

**Summary.** The fundamental limitation of passive retrieval lies in its inability to perform reasoning concurrently with memory access, leading to three weaknesses: (i) an inability to revise strategies based on intermediate state, such as identifying 'July' as a temporal anchor; (ii) the accumulation of noise due to fixed aggregation ; and (iii) heavy reliance on pre-constructed structures, limiting flexibility and scalability. A detailed analysis is included in Appendix A.

### 2.3. Motivation from Cognitive Memory Systems

These limitations highlight the necessity of moving beyond static retrieval toward an active memory reconstruction paradigm, which is strongly supported by cognitive neuroscience research (Rugg & Renoult, 2025). As illustrated in Figure 3, studies in human memory suggest that recall unfolds sequentially: contextual cues trigger the reactivation of *engrams*, compact internal memory states formed from past experiences, whose activation in turn biases and constrains subsequent recall, progressively reconstructing a coherent memory (Rashid et al., 2016). Motivated by this perspective, we adopt a Cue–Tag–Content architecture,

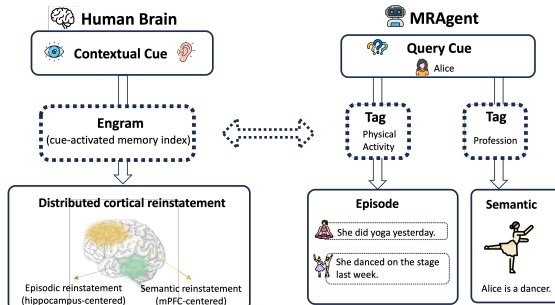

*Figure 3.* Functional correspondence between human memory reconstruction and the MRAgent architecture.

where tags serve as intermediate associative structures that encode how cues are linked to memory contents and guide the memory reconstruction process. Memory content can be distinguished into episodic memory for concrete events and semantic memory for shared concepts and knowledge, as supported by cognitive neuroscience (Manns et al., 2003).

## 3. Associative Memory System

To enable the active reconstruction paradigm described in Section 2, we organize the agent's memory as a heterogeneous graph. This section details the core *Cue–Tag–Content* architecture and its organization into multi-granular layers.

### 3.1. Cue–Tag–Content Associative Memory

Motivated by the need for active memory reconstruction in Section 2, we organize memory as a structured associative graph rather than a flat collection of retrievable items.

As illustrated in Figure 4 (a), the memory construction pipeline consists of two phases: element generation and graph construction. The first phase employs LLM to extract and generate memory elements from dialogs. Then, these elements are utilized to construct a memory graph.

**Unified Memory System.** We model the memory system as a heterogeneous graph $\mathcal{M} = (\mathcal{C}, \mathcal{V}, \mathcal{R})$. The graph contains two primary categories of nodes: (1) **Cues** $c \in \mathcal{C}$ are fine-grained keywords such as entities or attributes. (2) **Contents** $v \in \mathcal{V}$ store specific memory items.

Associations between cues and contents are encoded as typed relations: $\mathcal{R} \subseteq \mathcal{C} \times \mathcal{G} \times \mathcal{V}$, where each triple $(c, g, v) \in \mathcal{R}$ connects a cue $c$ to a content node $v$ through a relation attribute $g$, referred to as a **Tag**.

**Associative Tags.** Tags are utilized to summarize associative relations between fine-grained cues and content units. Based on these associations, the LLM performs a *two-stage retrieval* process, where it first selects a small set of relevant tags and then retrieves content conditioned on the selected tags. In large and complex memory graphs, naively

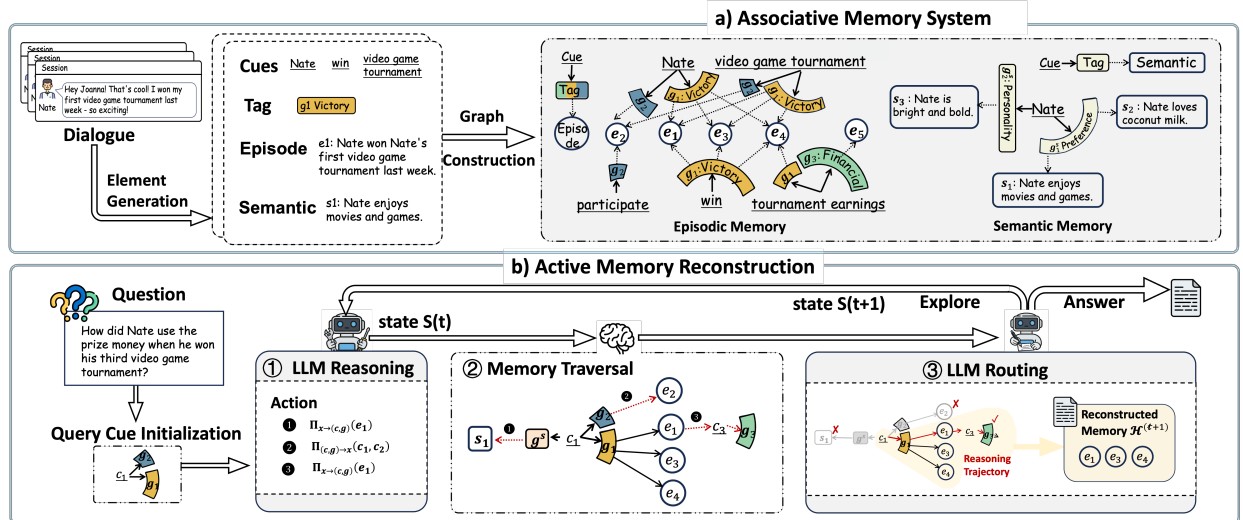

*Figure 4.* MRAgent: An Associative Memory System with LLM-Driven Active Memory Reconstruction. (a) MRAgent constructs an associative memory system from dialogues, organizing episodic and semantic memories through Cue–Tag–Content structures that explicitly encode semantic and relational associations through tags. (b) Upon a query, the agent performs active memory reconstruction, where an LLM iteratively reasons over cues and tags to traverse the memory graph and selectively reconstruct task-relevant memory.

expanding $n$-hop neighbors from content nodes often leads to combinatorial explosion and the inclusion of substantial irrelevant memories. By introducing tags as explicit associative intermediates, MRAgent enables guided and flexible reasoning over the memory graph. Tags provide semantic guidance, allowing the agent to evaluate and prune traversal branches to avoid incurring the computational cost of processing full episodic content.

To realize this two-stage retrieval process, we define two induced mapping operators over the memory relation $\mathcal{R}$. Specifically, $\phi_{c \to g}$ activates candidate associative tags from a given cue, while $\phi_{(c,g) \to v}$ retrieves content conditioned on both the cue and a selected tag:

$$\phi_{c \to g}(c) \triangleq \{g \mid (c, g, \cdot) \in \mathcal{R}\},$$
$$\phi_{(c,g) \to v}(c, g) \triangleq \{v \mid (c, g, v) \in \mathcal{R}\}. \quad (5)$$

Together, these operators decouple associative reasoning from content-level retrieval, making selective and reconstructive memory access tractable in large-scale graphs.

### 3.2. Multi-Granular Memory Layers

Inspired by human memory, we organize memory contents into complementary types spanning concrete events, stable knowledge, and higher-level abstractions. Episodic memory preserves event-specific information grounded in particular contexts, semantic memory captures abstract and relatively stable knowledge distilled across events, and topic-level abstractions summarize recurring patterns shared across episodes. Based on this distinction, we organize the memory graph into multiple functional layers, each supporting

different forms of reasoning and retrieval.

**Episodic Layer (Cue–Tag–Episode).** The episodic layer stores event-specific memory units $e_i \in \mathcal{V}^e$, each corresponding to a concrete experience occurring at a particular time. Episodes are retrievable through a set of fine-grained cues $\mathcal{C}_i \subseteq \mathcal{C}$ such as entities, actions, or contextual keywords and are routed by tags $g_i$ that summarize associative relations between cues and episodic content. To support temporal reasoning, episodic memories are further organized along a unified timeline, allowing temporal constraints to be imposed during reconstruction.

**Semantic Layer (Cue–Tag–Semantic).** The semantic layer captures relatively stable knowledge $s_i \in \mathcal{V}^s$, such as personal attributes, preferences, and general facts extracted from raw dialogue content. Each semantic node is anchored to a cue $c_i$, while the tag encode aspect-level associations of that cue, such as personality traits, long-term preferences, or factual attributes. This design enables direct access to targeted semantic information without requiring retrieval over potentially long episodic histories.

**Abstraction Layer (Topics).** The abstraction layer stores topic nodes $\tau \in \mathcal{V}^\tau$, each summarizing recurring patterns shared across a coherent set of episodes. Topics are connected to their constituent episodes and serve as a computational abstraction that supports efficient top-down transitions $\phi_{\tau \to e}$, allowing the agent to first localize a relevant topic and then descend to the associated episodes.

Together, these multi-granular memory layers allow MRAgent to flexibly combine event-level evidence from episodic memory, abstract knowledge from semantic memory, and

higher-level structure from topic abstractions, supporting both contextualized and conceptual reasoning.

### 3.3. Memory Population via LLM Distillation

The memory system is populated through an automated distillation pipeline applied to the input stream $T$. The stream is first segmented into episodic units $e_i$, each corresponding to a coherent event grounded in a specific context. For every episodic unit, tags and cues are extracted using LLM-based components:

$$g_i = F_{\text{LLM}}^{\text{tag}}(e_i), \qquad C_i = F_{\text{LLM}}^{\text{cue}}(e_i), \tag{6}$$

where $F_{\text{LLM}}^{\text{tag}}$ produces a short associative tag $g_i$ that summarizes the relational pattern of the episode, and $F_{\text{LLM}}^{\text{cue}}$ extracts a set of fine-grained cues $C_i$ such as entities, attributes, and salient descriptors. Each cue in $C_i$ is then linked to the episodic unit $e_i$ through the tag $g_i$, forming the Cue–Tag–Episode relations that constitute the episodic layer of the memory graph. Semantic units are extracted in a similar manner, yielding stable, abstract knowledge that persists across individual episodes and is anchored to entity-level cues through aspect-level tags.

To support reasoning at a coarser granularity, topic nodes are generated by summarizing the shared themes of related episodes, and each topic node is connected to its constituent episodes. This layered distillation organizes the input stream into a multi-granular associative structure, allowing the agent to access memory at the level of concrete events, stable facts, or higher-level topics as required during reconstruction. Additional details are provided in Appendix B.1.

## 4. MRAgent: Reconstructive Memory Agent

In this section, we present **MRAgent**, which formulates memory access as an active reconstruction process. We first define the reconstruction state and traversal actions (Section 4.1), and then describe the memory reconstruction process in detail (Section 4.2). Finally, we provide a theoretical analysis of its expressivity (Section 4.3).

### 4.1. Reconstructive Memory Framework

Rather than executing a predefined retrieval pipeline, MRAgent performs active reconstruction within a structured memory system. This process is formalized through an explicit reconstruction state and a set of traversal actions, which collectively characterize possible reconstruction trajectories.

**Reconstruction State.** MRAgent maintains an explicit reconstruction state that guides the selection of subsequent traversal directions during memory reconstruction. At step $t$, the reconstruction state is defined as

$$\mathcal{S}^{(t)} = (\mathcal{Z}^{(t)}, \mathcal{H}^{(t)}), \tag{7}$$

where $\mathcal{Z}^{(t)}$ denotes the *active set* of memory elements (including cues, tags and contents), serving as the candidates for the next traversal step, and $\mathcal{H}^{(t)}$ denotes the *reconstructed context* consisting of evidence accumulated in previous steps, which conditions subsequent traversal directions.

**Traversal Actions.** We define a finite set of traversal actions $\mathcal{A} = \{\Pi_1, \ldots, \Pi_m\}$ to specify how the LLM explores the structured memory graph during reconstruction. Each traversal action is induced by a predefined mapping operator $\phi$ introduced in Eq. (5), including Cue→Tag, (Cue,Tag)→Content, and Content→(Cue,Tag).

*Forward traversal* actions expands the active set along Cue–Tag–Content relations, enabling the agent to retrieve new memory contents conditioned on the current cues and tags. Specifically, $\Pi_{c \to g}$ activates associative tags from a given set of cues $\mathcal{C}^{(t)}$, and $\Pi_{(c,g) \to v}$ retrieves memory content jointly conditioned on selected cues and tags $(\mathcal{C}^{(t)}, \mathcal{G}^{(t)})$:

$$\Pi_{c \to g}(\mathcal{C}^{(t)}) \triangleq \bigcup_{c' \in \mathcal{C}^{(t)}} \phi_{c \to g}(c'),$$

$$\Pi_{(c,g) \to v}(\mathcal{C}^{(t)}, \mathcal{G}^{(t)}) \triangleq \bigcup_{c' \in \mathcal{C}^{(t)}} \bigcup_{g' \in \mathcal{G}^{(t)}} \phi_{(c,g) \to v}(c', g'). \tag{8}$$

*Reverse traversal* actions allows retrieved content to activate new cues and tags for subsequent exploration, enabling the agent to refine or redirect its reconstruction trajectory based on intermediate evidence. Formally, $\Pi_{v \to (c,g)}$ identifies cues and tags associated with the retrieved content:

$$\Pi_{v \to (c,g)}(\mathcal{V}^{(t)}) \triangleq \{(c', g') \mid \exists v' \in \mathcal{V}^{(t)}, \ (c', g', v') \in \mathcal{R}\}, \tag{9}$$

Together, these forward and reverse traversal actions define a compositional action space that allows the LLM to expand and refine memory traversal during reconstruction.

### 4.2. Memory Reconstruction Process

With the reconstruction state and traversal actions defined, we detail the memory reconstruction process, which iteratively explores relevant memory contents through state updates and traversal actions. Given a query, MRAgent first extracts a set of fine-grained cues and matches them against the stored cue set, obtaining the initial active set $\mathcal{Z}^{(0)}$ and the initial reconstruction state $\mathcal{S}^{(0)} = (\mathcal{Z}^{(0)}, \varnothing)$. Starting from this state, MRAgent starts an iterative reconstruction loop consisting of LLM reasoning, controlled memory traversal, and LLM-guided routing.

**LLM Reasoning and Action Selection.** Conditioned on reconstruction state, the LLM reasons over the query $x$, the accumulated context $\mathcal{H}^{(t)}$ to select a set of traversal actions

$\mathcal{A}^{(t)} \subseteq \mathcal{A}$ based on current active set $\mathcal{Z}^{(t)}$:

$$\mathcal{A}^{(t)} = f_{\text{select}}\big(x, \mathcal{H}^{(t)}, \mathcal{Z}^{(t)}\big), \qquad (10)$$

where $f_{\text{select}}$ denotes the LLM-based action-selection function that selects promising directions for expansion, thereby reducing noise. Furthermore, conditioning on the accumulated evidence $\mathcal{H}^{(t)}$ enables the agent to discover new cues and dynamically adjust its reasoning trajectory.

**Controlled Memory Traversal.** For each selected traversal action $\Pi_a \in \mathcal{A}^{(t)}$, the memory system executes the corresponding traversal operator to generate candidate nodes:

$$\widetilde{\mathcal{Z}}^{(t+1)} = \bigcup_{a \in \mathcal{A}^{(t)}} \Pi_a\big(\mathcal{Z}^{(t)}\big), \qquad (11)$$

where $\widetilde{\mathcal{Z}}^{(t+1)}$ is the new candidate set. This step expands the traversal trajectories, guided by LLM-selected actions rather than exhaustive graph expansion.

**LLM Routing and State Update.** Based on the generated candidates, the LLM selects the most relevant content and prunes irrelevant branches to ensure the accumulated context remains concise and focused. Formally, the next active set and reconstruction state are updated as follows:

$$\begin{aligned} \mathcal{Z}^{(t+1)} &= f_{\text{route}}\big(x, \mathcal{H}^{(t)}, \widetilde{\mathcal{Z}}^{(t+1)}\big), \\ \mathcal{H}^{(t+1)} &= \mathcal{H}^{(t)} \cup \mathcal{Z}^{(t+1)}, \end{aligned} \qquad (12)$$

where $f_{\text{route}}$ is the LLM-based routing function that evaluates semantic associations between the query, the reconstructed context, and newly retrieved memory contents. Unlike surface-level matching or predefined traversal, this routing step incorporates semantic associations and structural relations exposed by the memory graph. Following the state update, the accumulated context $\mathcal{H}^{(t+1)}$ is evaluated by $\mathcal{C}_{\text{LLM}}$ to determine if the evidence is sufficient to answer the query or if further exploration is required.

Through this process, MRAgent integrates LLM reasoning directly into the multi-turn memory reconstruction. Selective evidence expansion reduces noise and improves efficiency, while conditioning on intermediate evidence enables flexible adjustment of the reasoning trajectory. Implementation details and algorithms are provided in Appendix B.2.

### 4.3. Theoretical Analysis: Active vs. Passive Retrieval

MRAgent is designed as an *active* retrieval approach over a graph-based memory. In this section, we formalize the advantage of such active retrieval over passive retrieval, from an approximation-theoretic view. For generality and notational simplicity, our theory generalizes from the (cue, tag, episode) formulation in MRAgent to arbitrary heterogeneous graphs with textual information on each node.

Given retrieval budget $T$, recall that *active reconstruction* adaptively chooses the next node to retrieve based on what it has already retrieved, while a *passive* retriever must commit to all $T$ retrievals upfront as a function of the query alone.

These retrieval strategies induce different *hypothesis classes* (sets of input–output mappings). Specifically, let $\mathcal{H}_{\text{active}}^{\text{LM}}(T)$ contain all predictors implementable by an LM that can make $T$ adaptive retrieval calls to the graph, and let $\mathcal{H}_{\text{passive}}^{\text{LM}}(T)$ contain those implementable when the $T$ retrieval calls are fixed in advance (non-adaptive). Then our main result is:

**Theorem 4.1** (Active retrieval is strictly more powerful than passive retrieval)**.** *For any retrieval budget $T \geq 2$, the passive hypothesis class is strictly contained in the active hypothesis class:*

$$\mathcal{H}_{\text{passive}}^{\text{LM}}(T) \subsetneq \mathcal{H}_{\text{active}}^{\text{LM}}(T).$$

Intuitively, LMs with active retrieval can learn any function that LMs with passive retrieval can, but not vice versa. The full statement and proofs are given in Appendix C.

## 5. Experiments

In this section, we conduct comprehensive experiments to evaluate MRAgent and answer the following research questions: (RQ1) How does MRAgent perform compared to existing memory systems? (RQ2) How does MRAgent compare to baselines in terms of computational cost? (RQ3) How do different components of MRAgent contribute to overall performance? (RQ4) How does multi-turn reasoning progressively improve memory reconstruction? (RQ5) How does MRAgent behave in real conversational scenarios?

### 5.1. Experiment Setup

**Benchmarks.** We evaluate MRAgent on two widely used benchmarks for long-context memory evaluation: (1) *LoCoMo* (Maharana et al., 2024), which focuses on long conversational memory understanding, and (2) *Long-MemEval* (Wu et al., 2025), which is designed to assess long-term memory system across multiple sessions with longer interaction histories per query.

**Baselines.** We evaluate MRAgent against representative memory-augmented baselines: Retrieval-augmented generation (RAG), *LangMem* (LangChain, 2025), *A-Mem* (Xu et al., 2025), *MemoryOS* (Kang et al., 2025), and *Mem0* (Chhikara et al., 2025).

**Implementation.** We evaluate all methods using two LLM backbones: Gemini-2.5-Flash and Claude-Sonnet-4.5. Following prior work, we report $F_1$ and LLM-Judge (J) scores using GPT-4o-mini, and additionally report evidence recall (Recall) for analysis.

*Table 1.* Performance across different question types on LoCoMo. Evaluation metrics include F1 score (F1), and LLM-Judge score (J). All values are reported as percentages. **Bold** indicates the best result, and underline indicates the second best.

| Model | Method | Multi-hop | | Temporal | | Open Domain | | Single hop | | Overall |
|-------|--------|-----------|-----|----------|-----|-------------|-----|-----------|-----|---------|
| | | $F_1 \uparrow$ | $J \uparrow$ | $F_1 \uparrow$ | $J \uparrow$ | $F_1 \uparrow$ | $J \uparrow$ | $F_1 \uparrow$ | $J \uparrow$ | $J \uparrow$ |
| Gemini | RAG | 34.89 | $58.16_{\pm 0.45}$ | 43.52 | $49.22_{\pm 0.15}$ | 25.68 | $41.67_{\pm 0.47}$ | 53.69 | $69.20_{\pm 0.05}$ | 61.30 |
| | A-Mem | 33.81 | $53.54_{\pm 0.33}$ | 40.22 | $49.53_{\pm 0.44}$ | 12.49 | $33.33_{\pm 0.49}$ | 46.39 | $61.83_{\pm 0.10}$ | 55.97 |
| | MemoryOS | 41.42 | $63.82_{\pm 0.44}$ | 35.91 | $47.04_{\pm 0.53}$ | 23.43 | $41.66_{\pm 0.85}$ | 54.82 | $71.90_{\pm 0.22}$ | 63.35 |
| | LangMem | 40.67 | $61.34_{\pm 0.79}$ | 44.70 | $53.58_{\pm 0.25}$ | 20.49 | $38.54_{\pm 0.33}$ | 48.20 | $69.68_{\pm 0.12}$ | 62.86 |
| | Mem0 | **45.17** | $68.79_{\pm 1.21}$ | 58.19 | $61.68_{\pm 0.29}$ | 26.24 | $41.66_{\pm 1.63}$ | 54.37 | $73.72_{\pm 0.10}$ | 68.31 |
| | **MRAgent** | 43.69 | $\mathbf{75.17}_{\pm 0.33}$ | **67.66** | $\mathbf{80.37}_{\pm 0.15}$ | **32.51** | $\mathbf{68.75}_{\pm 0.98}$ | **64.08** | $\mathbf{90.48}_{\pm 0.10}$ | **84.21** |
| Claude | RAG | 34.53 | $57.45_{\pm 0.59}$ | 43.39 | $48.29_{\pm 0.15}$ | 26.56 | $43.75_{\pm 0.34}$ | 53.66 | $69.20_{\pm 0.06}$ | 61.10 |
| | A-Mem | 42.45 | $71.67_{\pm 0.22}$ | 47.73 | $55.48_{\pm 0.28}$ | 22.02 | $47.57_{\pm 0.46}$ | 55.19 | $74.71_{\pm 0.05}$ | 68.45 |
| | MemoryOS | 32.94 | $60.99_{\pm 0.44}$ | 39.14 | $51.09_{\pm 0.29}$ | 18.29 | $48.95_{\pm 0.15}$ | 45.46 | $66.49_{\pm 0.21}$ | 61.18 |
| | LangMem | 44.37 | $70.92_{\pm 0.25}$ | 56.64 | $80.68_{\pm 0.36}$ | 22.66 | $54.71_{\pm 0.55}$ | 54.36 | $83.12_{\pm 0.15}$ | 78.61 |
| | Mem0 | 48.66 | $75.88_{\pm 0.67}$ | 49.50 | $53.58_{\pm 0.44}$ | 28.58 | $56.25_{\pm 0.49}$ | 54.43 | $74.07_{\pm 0.06}$ | 69.02 |
| | **MRAgent** | **56.72** | $\mathbf{90.19}_{\pm 0.29}$ | **69.82** | $\mathbf{85.34}_{\pm 0.25}$ | **34.67** | $\mathbf{71.57}_{\pm 0.10}$ | **68.62** | $\mathbf{91.10}_{\pm 0.15}$ | **88.32** |

*Table 2.* Performance on LONGMEMEVAL evaluated by LLM-Judge $\uparrow$. Best and second-best results among Gemini-backbone methods are marked in **bold** and underline, respectively. Mul.: multi-session; Sgl.: single-session-user; Tmp.: temporal-reasoning; Pref.: single-session-preference. MRAgent* uses Claude for retrieval while memories are constructed by Gemini. The complete table is provided in Appendix D.5.

| Method | Mul. | Sgl. | Tmp. | Pref. | Overall |
|--------|------|------|------|-------|---------|
| RAG | 54.89 | 85.71 | 42.86 | 33.33 | 54.65 |
| A-Mem | 42.85 | 90.00 | 45.11 | 46.43 | 52.98 |
| MemoryOS | 56.39 | 87.14 | 38.35 | 46.67 | 54.92 |
| LangMem | 52.63 | 78.57 | 45.71 | 36.67 | 53.77 |
| Mem0 | 50.38 | 78.57 | 45.11 | 40.00 | 53.01 |
| MRAgent | **68.42** | **92.85** | **68.42** | **66.67** | **72.95** |
| MRAgent* | 86.46 | 92.85 | 85.71 | 78.57 | 86.76 |

Additional details are reported in the Appendix D.

## 5.2. Main Results (RQ1)

**MRAgent consistently outperforms all baselines across question types and backbones.** As shown in Table 1, MRAgent improves the overall $J$ score from $68.31$ to $84.21$ under the Gemini backbone (an $23.3\%$ relative gain), and achieves a $12.4\%$ improvement under the Claude backbone.

These gains can be attributed to two key factors. *First*, the Cue–Tag–Content (CTC) memory design explicitly encodes associative relations via tags, integrating semantic information into the memory graph structure. Compared with graph-based retrieval methods that expand purely along pre-defined structural links, the CTC memory allows MRAgent to select retrieval directions based on semantic relevance, thereby improving its ability to locate supporting evidence. *Second*, MRAgent integrates LLM reasoning into multi-turn memory access, allowing retrieval directions to adapt to accumulated evidence and to progressively form coherent

*Table 3.* Per-sample token consumption and runtime on long-term LONGMEMEVAL across different memory systems using the Gemini backbone. Results include both memory construction and retrieval. **Bold** indicates the best result, and underline indicates the second best.

| Method | Token Consumption | Runtime(s) |
|--------|-------------------|------------|
| A-Mem | 632k | 1,122.23 |
| MemoryOS | 273k | 3,135.54 |
| LangMem | 3,268k | 1,209.57 |
| Mem0 | 245k | **533.29** |
| **MRAgent** | **118k** | 586.11 |

reasoning chains. This adaptive reconstruction process enables the agent to focus on relevant evidence and follow evidence-guided retrieval paths, leading to superior performance over passive retrieval baselines.

Table 2 further shows consistent improvements on LONG-MEMEVAL, achieving a relative improvement of $32\%$ over the strongest baseline.

## 5.3. Cost Analysis (RQ2)

Table 3 reports the total token consumption and runtime per sample on LONGMEMEVAL, which is designed for long-term memory assess across multiple dialogue sessions.

**MRAgent improves information efficiency through selective, on-demand memory access.** MRAgent reduces prompt tokens to 118k, a significant decrease from baselines like A-Mem (632k). Unlike existing methods that repeatedly summarize history and analyze intricate dependencies during construction, MRAgent maintains a lightweight construction phase and defers complex relation-building to the retrieval stage, where it is performed in a query-specific manner. Moreover, by leveraging associative tags to semantically guide retrieval directions, the agent can prune

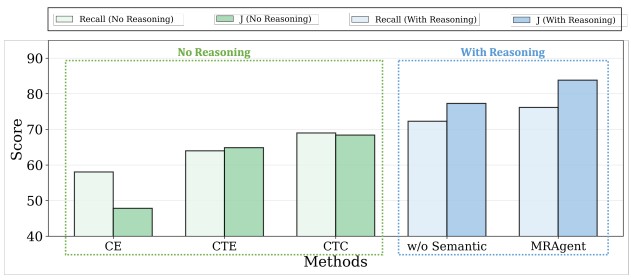

*Figure 5.* Ablation results on LOCOMO for multi-hop queries, evaluated using Recall and LLM-Judge (J) under Claude backbone.

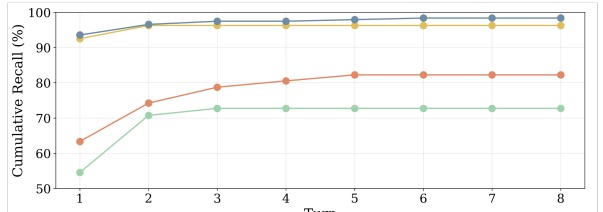

(a) Cumulative Evidence Recall as a function of reasoning turns across question types.

|  | **Multi-hop** | **Temporal** | **Open Domain** | **Single hop** |
|---|---|---|---|---|
| **Average Turns** | 3.16 | 2.42 | 2.60 | 2.07 |
| **Max Valid Turns** | 2.65 | 2.40 | 1.09 | 1.28 |

(b) Average and valid reasoning turns required for different question types.

*Figure 6.* Analysis of multi-turn reasoning on LOCOMO under the Claude backbone.

irrelevant paths before accessing expensive episodic content. This "on-demand" approach ensures that computational resources are focused strictly on query-relevant evidence.

### 5.4. Ablation Study (RQ3)

Figure 5 presents ablation results on multi-hop questions in LOCOMO, isolating the contributions of memory structure and reasoning mechanisms. We consider three structural variants: CE (Cue→Episode) with direct indexing, CTE (Cue–Tag–Episode) with mediated episodic retrieval, and CTC (Cue–Tag–Content) with the full memory structure. The CTE and CTC variant is evaluated both without reasoning (green bars) and with reasoning (blue bars).

**Active multi-step reasoning is a primary factor underlying the observed performance gains.** Across all memory structures, variants with reasoning (blue bars) consistently outperform their structure-only counterparts (green bars). This indicates that multi-step reasoning and traversal are crucial for accumulating evidence and supporting multi-hop inference, whereas one-shot retrieval is insufficient to resolve complex multi-hop queries.

**Associative tags provide effective semantic guidance for retrieval.** Within the no-reasoning setting (green bars), performance improves monotonically from CE to CTE to CTC, demonstrating that richer associative structures enable more reliable retrieval. In particular, tags help guide retrieval toward semantically relevant directions and reduce the inclusion of fragmented or irrelevant memory units.

**Episodic and semantic memory layers are complementary.** Removing the semantic memory component leads to a clear performance degradation (blue bars). While episodic memory preserves event-specific details, semantic memory captures stable, abstract knowledge that is essential for multi-hop reasoning.

### 5.5. Multi-turn Reasoning Analysis (RQ4)

As shown in Figure 6 (a), we evaluate cumulative evidence recall across reasoning turns on the LOCOMO dataset. Figure 6 (b) summarizes the average number of reasoning turns

to convergence (Average Turns) and the maximum number of turns that retrieve valid information (Max Valid Turns).

**Multi-turn reasoning progressively recovers missing evidence.** As shown in Figure 6 (a), single-hop (blue line) and temporal (yellow line) queries reach near-perfect recall within approximately three turns, whereas multi-hop queries benefit substantially from iterative exploration, with recall improving by over 30% across successive steps (red line). This highlights the necessity of multi-step reconstruction for resolving compositional and long-range dependencies.

**The agent autonomously evaluates accumulated context to guide search and termination.** As shown in Figure 6 (b), Max Valid Turns closely matches Average Turns, suggesting that the LLM effectively determines when to continue searching and when to stop. This behavior minimizes redundant exploration. Furthermore, experiments in Appendix D.6 show that increasing the parallel retrieval budget cannot substitute for deeper reconstruction depth.

### 5.6. Case Study (RQ5)

We include a qualitative case study in Figure 7 demonstrating MRAgent's ability to actively reconstruct multi-session evidence over a structured memory graph for complex, temporally grounded queries, with detailed analysis provided in the Appendix D.8.

## 6. Related Work

**Retrieval-Augmented Generation.** RAG (Lewis et al., 2020) augments LLMs by injecting external documents into prompts via similarity search, where memory is an unstructured vector store and retrieval reduces to a one-shot query-time top-$k$ selection. Two variants extend this paradigm. GraphRAG (Han et al., 2025) organizes the retrieval corpus into graph structures and retrieves through community summaries and neighborhood expansion, improving global and multi-hop reasoning over flat similarity search. Agen-

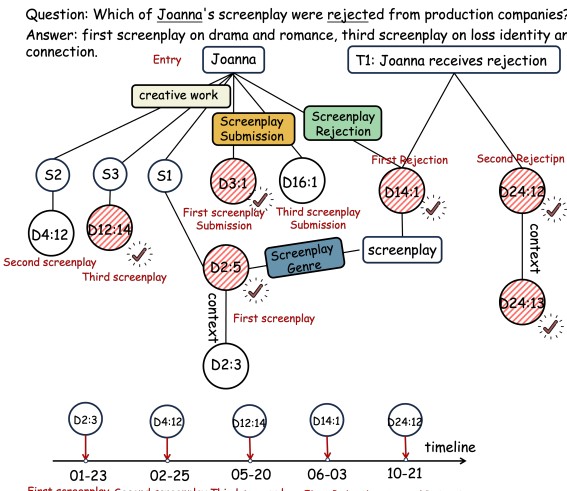

Question: Which of Joanna's screenplay were rejected from production companies?
Answer: first screenplay on drama and romance, third screenplay on loss identity and connection.

*Figure 7.* Reasoning trajectory of MRAgent over the memory graph for a multi-session query. Starting from the cue "Jonna", the agent traverses multiple associative tags to retrieve both episodic memories (e.g., screenplay submissions) and semantic information (e.g., background about the screenplay). It then reasons over higher-level topics to recover rejection events and aligns them with previously retrieved submissions, enabling a coherent answer to the query. Here, $Dx:x$ denotes concrete event instances.

tic RAG instead moves retrieval inside the reasoning loop. Search-o1 (Li et al., 2025) couples large reasoning models with an agentic mechanism that issues search queries on demand whenever a knowledge gap is detected and refines retrieved documents before integrating them into the reasoning chain, and Search-R1 (Jin et al., 2025) trains this behavior through reinforcement learning over multi-turn query generation. These methods retrieve from open or external corpora to fill factual knowledge gaps within a single reasoning task, rather than reconstructing over an agent's own persistent interaction history.

**Graph-based Memory.** Graph-based memory systems organize agent memory as a graph to capture structural dependencies among memory units. A-Mem (Xu et al., 2025) constructs structured memory notes linked through LLM-assisted relation extraction and retrieves via seed selection followed by neighborhood expansion. Zep (Rasmussen et al., 2025) maintains a bi-temporal knowledge graph that tracks when facts hold and invalidates outdated edges, supporting retrieval over evolving knowledge. LiCoMemory (Huang et al., 2025) organizes memory as a lightweight hierarchical graph that uses entities and relations as a semantic indexing layer, with temporal and hierarchy-aware search for efficient retrieval. While these representations improve relational and multi-hop access, traversal is governed by predefined operators and does not adapt to evidence accumulated during retrieval.

**Hierarchical and Persistent Memory Systems.** Hierar-

chical memory systems maintain long-lived memory that is continually updated across interactions. MemoryOS (Kang et al., 2025) organizes memory into a short, mid, and long-term hierarchy for structured access. Mem0 (Chhikara et al., 2025) maintains a compact set of salient facts through LLM-driven add, update, and delete operations. SeCom (Pan et al., 2025) constructs the memory bank at the level of topically coherent segments for more accurate retrieval. These systems advance how memory is stored and updated over time, but their retrieval procedures remain passive, selecting memory units as a fixed function of the query without reasoning over intermediate evidence.

## 7. Conclusion and Discussion

We proposed MRAgent, a reconstructive memory agent that formulates memory access as an active, multi-step reconstruction process over a structured memory graph. A key design choice of MRAgent is to shift the modeling of complex relational dependencies to the retrieval stage, enabling the agent to resolve more complex queries with fewer computational cost through targeted, state-dependent exploration. As a result, our current implementation adopts a relatively simple memory construction strategy, without introducing additional complexity in memory updating or forgetting mechanisms. This design choice also exposes several limitations that suggest directions for future work. First, because relational reasoning is deferred to retrieval, the cost of reconstruction grows with the depth of exploration, and queries that require many traversal steps incur higher latency than single-shot retrieval. Second, our static construction does not update or consolidate memory over time, so the memory graph grows monotonically as interactions accumulate, raising storage overhead in long-lived deployments. Addressing these limitations through adaptive construction, lightweight memory maintenance, and more robust traversal policies is a promising avenue for extending active reconstruction to broader long-horizon settings.

## Acknowledgements

This research is supported by the Ministry of Education, Singapore, under the Academic Research Fund Tier 2 (FY2025) (Grant T2EP20124-0038).

## Impact Statement

This work aims to advance the design of memory systems for large language model (LLM) agents by introducing an active, reconstructive paradigm for long-horizon memory reasoning. Potential positive impacts include more reliable and context-aware AI assistants for applications such as personal assistants, decision support systems, and long-term human–AI interaction, where accurate recall and reasoning

over past information are essential. At the same time, as with other memory-augmented AI systems, the ability to store and reason over long-term interaction data raises considerations around privacy, data governance, and responsible deployment. These concerns are not specific to our method but apply broadly to persistent memory in AI agents.

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

# A. Detailed Analysis of Passive Retrieval

This appendix provides a unified analysis of the retrieval mechanisms in representative memory systems for LLM agents, aligning all methods with the formulation in Section 2. Recall that memory access operates over an external memory $\mathcal{M}$ with units $\mathcal{V} = \{v_1, \ldots, v_N\}$. Given a query $x$, passive retrieval returns a fixed set $\pi_{\mathrm{p}}(x)$, while active reconstruction selects units sequentially via a stateful policy $v^{(t)} = \pi_{\mathrm{a}}^{(t)}(x, S^{(t-1)})$.

**Retrieval-Augmented Generation.** Retrieval-augmented generation (RAG) adopts a classic similarity-based retrieval paradigm, where an external memory is organized as a vector store and queried in a single shot. Given a query $x$, RAG computes a relevance score between $x$ and each memory unit $v \in \mathcal{V}$, and retrieves the top-$k$ most similar items:

$$\pi_{\mathrm{p}}^{\mathrm{RAG}}(x) = \mathrm{TopK}(\{\mathrm{sim}(x, v)\}_{v \in \mathcal{V}}, k). \tag{13}$$

**A-Mem** (Xu et al., 2025). A-Mem introduces a graph structure over memory units, where nodes correspond to structured memory notes and edges encode semantic relations constructed during memory construction. Given a query $x$, A-Mem performs retrieval by first selecting a seed memory item via similarity, and then expanding its predefined neighborhood along the memory graph:

$$\mathcal{V}^* = \mathrm{TopK}(\{\mathrm{sim}(x, v)\}_{v \in \mathcal{V}}, k),$$
$$\pi_{\mathrm{p}}^{\text{A-MEM}}(x) = \mathcal{V}^* \cup \mathrm{Neighbor}(\mathcal{V}^*). \tag{14}$$

where $\mathrm{Neighbor}(\mathcal{V}^*) = \bigcup_{v \in \mathcal{V}^*} \mathrm{Neighbor}(v)$ returns the nodes directly connected to the seed set in the memory graph.

**MemoryOS** (Kang et al., 2025). MemoryOS organizes agent memory into a three-tier hierarchy with short-term memory (STM), mid-term memory (MTM), and long-term personal memories (LPM). Given a query $x$, it performs hierarchical retrieval by incorporating recent short-term context, relevant mid-term topic memories, and long-term persona information:

$$R_{\mathrm{MTM}}(x) = \mathrm{TopK}(\{\mathcal{F}_{score}(x, v)\}_{v \in \mathcal{V}_{\mathrm{MTM}}}), \quad R_{\mathrm{LPM}}^{\mathrm{fact}}(x) = \mathrm{TopK}\left(\{\mathcal{F}_{score}(x, v)\}_{v \in \mathcal{V}_{\mathrm{LPM}}^{\mathrm{fact}}}\right),$$
$$\pi_{\mathrm{p}}^{\text{MEMOS}}(x) = \mathcal{V}_{\mathrm{STM}} \cup R_{\mathrm{MTM}}(x) \cup \mathcal{V}_{\mathrm{LPM}}^{\mathrm{profile}} \cup R_{\mathrm{LPM}}^{\mathrm{fact}}(x), \tag{15}$$

where $\mathcal{V}_{\mathrm{LPM}}^{\mathrm{profile}}$ represents persona information, and $\mathcal{V}_{\mathrm{LPM}}^{\mathrm{fact}}$ denotes query-conditioned long-term factual memory.

**LangMem** (LangChain, 2025). LangMem maintains conversational memory by compressing dialogue history into a set of memory summaries, which are stored in a vector-based memory store and injected into the model context during inference. Each memory unit corresponds to a summarized dialogue message represented by an embedding vector. Given a query $x$, LangMem retrieves relevant memories via similarity-based search:

$$\pi_{\mathrm{p}}^{\text{LANGMEM}}(x) = \mathcal{S}_{\mathrm{LLM}}(\mathrm{TopK}(\{\mathrm{sim}(x, v)\}_{v \in \mathcal{V}}, k), x). \tag{16}$$

where $\mathrm{Summarize}_{\mathrm{LLM}}(\cdot)$ denotes an LLM-based summarization function that compresses retrieved memory entries into a concise context representation.

**Mem0** (Chhikara et al., 2025). Mem0 maintains a persistent memory store composed of natural-language facts extracted incrementally from conversations via an LLM-driven memory construction and update process. Each memory item corresponds to a salient factual statement, and memory evolution is governed by LLM-selected operations (ADD, UPDATE, DELETE, NOOP) to ensure consistency over time. Given a query $x$, Mem0 performs retrieval by selecting the most relevant memory items via similarity search:

$$\pi_{\mathrm{p}}^{\text{MEM0}}(x) = \mathrm{TopK}(\{\mathrm{sim}(x, v)\}_{v \in \mathcal{V}}, k). \tag{17}$$

**Conclusion.** From the above analysis, existing memory systems primarily focus on the design of memory representations, with retrieval procedures tightly coupled to the underlying structure. These approaches can be broadly categorized as similarity-based retrieval and graph-based retrieval. Although some methods employ structured memory representations, their retrieval processes remain passive: traversal operators are predefined and fixed, and do not adapt based on intermediate evidence during retrieval.

*Table 4.* Memory toolkit providing operations for controlled memory traversal.

| Tool | Mapping Function | Description |
|---|---|---|
| `query_tag_events` | $\phi_{(c,g)\to e}(c, g)$ | Retrieve episodic events associated with a cue–tag pair. |
| `query_conversation_time` | $\phi_{e\to t}(v^e e)$ | Return the conversation timestamp of an episodic event. |
| `query_event_keywords` | $\phi_{e\to(c,g)}(e)$ | Extract associated cues and tags from an episodic event. |
| `query_event_context` | $\phi_{e\to \text{ctx}}(e)$ | Retrieve contextual text surrounding the episodic event. |
| `query_personal_information` | $\phi_{c^s\to g^s}(c^s)$ | Return semantic aspects associated with a person entity. |
| `query_personal_aspect` | $\phi_{(c^s,g^s)\to v^s}(c^s, g^s)$ | Retrieve semantic content for a (person, aspect) pair. |
| `query_topic_events` | $\phi_{\tau\to e}(\tau)$ | Retrieve episodic events associated with a topic node. |

## B. Implementation and Execution Details

### B.1. Memory Construction Pipeline

**Cue–Tag–Episode.** To instantiate elements of the memory graph and explicitly model their associative relations, we employ an LLM to extract the components required to construct Cue–Tag–Episode triplets. Given the raw dialogue text $T$, we first rewrite and refine it to resolve contextual dependencies and make cues explicit, and then segment the processed text into coherent episodic units:

$$\{e_i\} \leftarrow \mathcal{R}_{\text{LLM}}(T), \tag{18}$$

where $\mathcal{R}_{\text{LLM}}$ performs raw text processing, including pronoun resolution, temporal normalization, and episodic segmentation. For each episodic segment $e_i$, we generate an associative tag and a set of fine-grained cues using LLM-based extractors:

$$g_i \leftarrow \mathcal{T}_{\text{LLM}}(x_i), \quad \mathcal{C}_i \leftarrow \mathcal{K}_{\text{LLM}}(x_i), \tag{19}$$

where $\mathcal{T}_{\text{LLM}}$ produces a short and precise phrase summarizing the core semantic or relational pattern of the episode, and $\mathcal{K}_{\text{LLM}}$ extracts a set of cues $C_i$, including entities, attributes, and other salient descriptors. We then add nodes for $x_i$ and each $c \in C_i$, and associate them with the tag $g_i$ to form the Cue–Tag–Episode relations.

**Cue–Tag–Semantic.** Semantic memory complements episodic reconstruction by providing relatively stable abstractions beyond individual episodes (Anokhin et al., 2024; Hu et al., 2025). Each semantic unit is represented as $(c_i^s, g_i^s, s_i)$, where $s_i$ denotes the semantic content, $c_i^s$ is an entity-level cue, and $g_i^s$ is an aspect-level tag. We extract such semantic units from the input text $T$ via an LLM-based semantic extraction function:

$$\{(c_i^s, g_i^s, s_i)\} \leftarrow \mathcal{S}_{\text{LLM}}(T), \tag{20}$$

where $\mathcal{S}_{\text{LLM}}$ identifies stable facts and attributes expressed in $T$, summarizes them into semantic content $s_i$, and assigns an entity-level cue $c_i^s$ together with an aspect-level tag $g_i^s$. The extracted $(c_i^s, g_i^s, s_i)$ units are inserted into the same memory graph and linked by Cue–Tag–Semantic associations.

**High-level Topic Memory.** While Tag captures episode-level semantic patterns, Topic summarizes recurrent patterns across episodes and provides a higher-level organization for multi-granular reasoning. This design is consistent with the view that higher-order structures can be abstracted from repeated episodic experiences (Pan et al., 2025; Tan et al., 2025). Concretely, we obtain topic nodes by applying an LLM-based abstraction function to the set of episodic segments:

$$\{\tau_j\} \leftarrow \mathcal{A}_{\text{LLM}}(\{e_i\}), \tag{21}$$

where each topic node $\tau_j$ summarizes a coherent set of episodes that share recurring semantic patterns. We then add topic–episode links to connect each $\tau_j$ with its associated episodes, enabling the agent to reason at a higher level when fine-grained episodic traversal is unnecessary.

### B.2. Executable Memory Reconstruction

Figure 8 presents a detailed illustration of the memory reconstruction process, accompanied by an example diagram and pseudocode. To realize multi-step reasoning trajectories, we design a specialized toolkit that executes the traversal actions determined by the LLM and returns newly retrieved evidence from the memory graph.

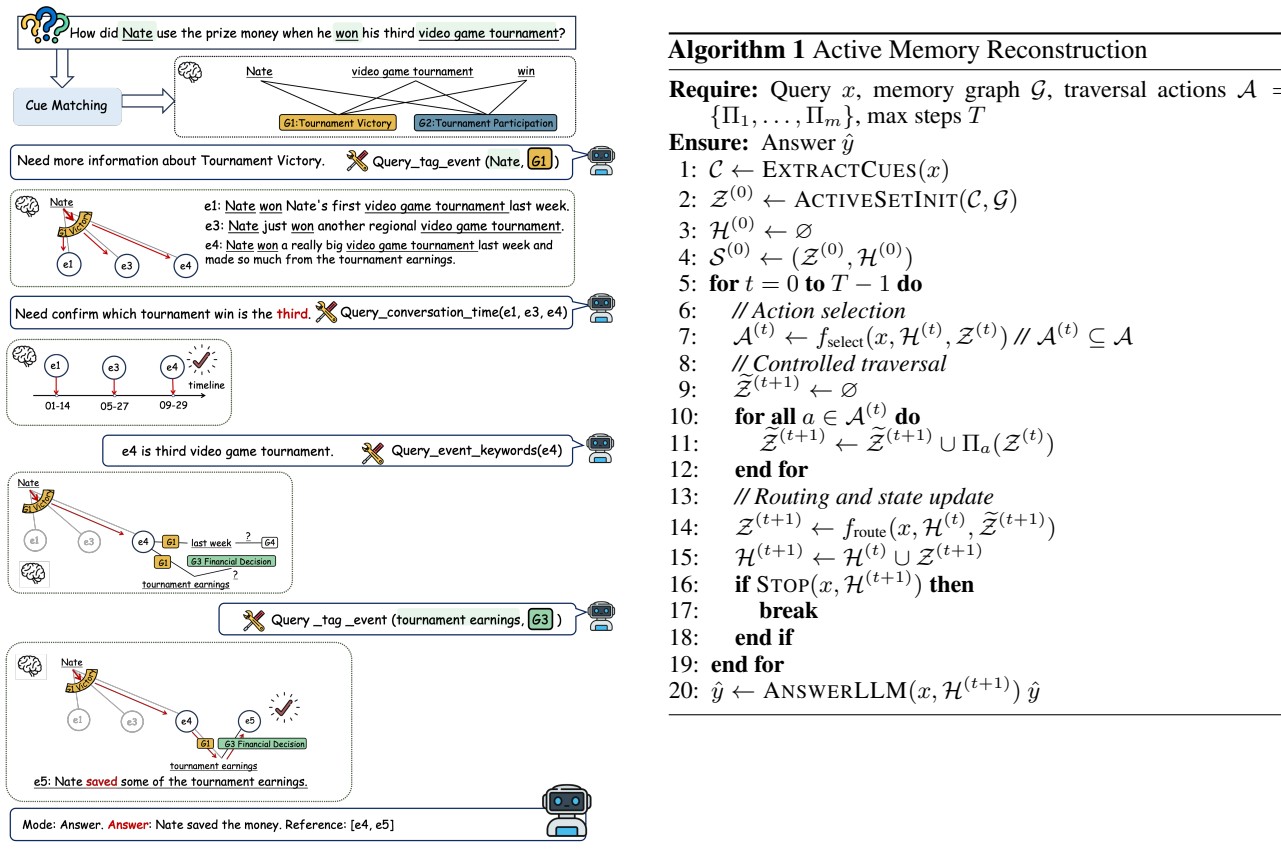

**Algorithm 1** Active Memory Reconstruction

**Require:** Query $x$, memory graph $\mathcal{G}$, traversal actions $\mathcal{A} = \{\Pi_1, \ldots, \Pi_m\}$, max steps $T$
**Ensure:** Answer $\hat{y}$
1: $\mathcal{C} \leftarrow \text{EXTRACTCUES}(x)$
2: $\mathcal{Z}^{(0)} \leftarrow \text{ACTIVESETINIT}(\mathcal{C}, \mathcal{G})$
3: $\mathcal{H}^{(0)} \leftarrow \varnothing$
4: $\mathcal{S}^{(0)} \leftarrow (\mathcal{Z}^{(0)}, \mathcal{H}^{(0)})$
5: **for** $t = 0$ **to** $T - 1$ **do**
6:     *// Action selection*
7:     $\mathcal{A}^{(t)} \leftarrow f_{\text{select}}(x, \mathcal{H}^{(t)}, \mathcal{Z}^{(t)})$ *// $\mathcal{A}^{(t)} \subseteq \mathcal{A}$*
8:     *// Controlled traversal*
9:     $\widetilde{\mathcal{Z}}^{(t+1)} \leftarrow \varnothing$
10:    **for all** $a \in \mathcal{A}^{(t)}$ **do**
11:      $\widetilde{\mathcal{Z}}^{(t+1)} \leftarrow \widetilde{\mathcal{Z}}^{(t+1)} \cup \Pi_a(\mathcal{Z}^{(t)})$
12:    **end for**
13:    *// Routing and state update*
14:    $\mathcal{Z}^{(t+1)} \leftarrow f_{\text{route}}(x, \mathcal{H}^{(t)}, \widetilde{\mathcal{Z}}^{(t+1)})$
15:    $\mathcal{H}^{(t+1)} \leftarrow \mathcal{H}^{(t)} \cup \mathcal{Z}^{(t+1)}$
16:    **if** $\text{STOP}(x, \mathcal{H}^{(t+1)})$ **then**
17:      **break**
18:    **end if**
19: **end for**
20: $\hat{y} \leftarrow \text{ANSWERLLM}(x, \mathcal{H}^{(t+1)}) \, \hat{y}$

*Figure 8.* Left: An illustrative example of memory reconstruction given a query. Right: Pseudocode of the memory reconstruction algorithm.

MRAgent operates under two execution modes, $\Psi \in \{\texttt{Navigate}, \texttt{Answer}\}$. In the $\texttt{Navigate}$ mode, the agent invokes the toolkit to progressively explore the memory graph and accumulate evidence. Once sufficient evidence has been collected, the agent transitions into the $\texttt{Answer}$ mode to generate the final response conditioned on the reconstructed context.

Table 4 summarizes the toolkit used for memory traversal and evidence acquisition. Each tool corresponds to a typed mapping between memory components, enabling the LLM to explicitly control the direction and granularity of memory access. Rather than retrieving a fixed set of memories, the agent composes these tools into sequential or parallel invocations to progressively reconstruct relevant context. At each navigation step, the LLM selects and invokes one or more tools based on the current reconstruction state. The returned results are treated as candidate evidence and are evaluated by the LLM to decide whether to expand, refine, or terminate a traversal path. This explicit tool-based interface ensures that memory access remains interpretable and controllable, while allowing the agent to adapt its retrieval strategy to intermediate evidence.

## C. Theoretical Analysis: Active vs. Passive Retrieval over a Heterogeneous KG Memory

MRAgent is designed as an *active* retrieval approach over a graph-based memory. This section formalizes an approximation-theoretic advantage of such *active* (adaptive) retrieval over *passive* (non-adaptive) retrieval.

### C.1. Setup

We consider tasks involving queries $x \in \mathcal{X}$, with answers $y$ from a label space $\mathcal{Y}$.

Our MRAgent method uses a heterogeneous knowledge graph (KG) memory with cues, tags and episodes. However, for generality and notational simplicity, our theory generalizes this to arbitrary heterogeneous KGs with textual information on each node.

**Heterogeneous KG memory.** For each problem size $n$, a memory $M \in \mathcal{M}_n$ is a heterogeneous KG with node set

$$\mathcal{V}(M) = \{v_1, v_2, \ldots, v_n\}.$$

Each node $v \in \mathcal{V}(M)$ has a type $\tau(v)$ in some type set, and a textual *payload* $p(v) \in \mathcal{P}$. The KG also has a finite set of relation labels $\mathcal{R}$ (e.g., tag types).

**Node-retrieval operator.** Our LM interacts with the KG via a *retrieval operator* that, given a node, returns both (i) the node payload and (ii) a bounded list of outgoing neighbors.

Formally, fix a branching bound $B \geq 1$. For each memory $M$, define

$$\mathrm{Retrieve}^M : \mathcal{V}(M) \to \mathcal{P} \times (\mathcal{R} \times \mathcal{V}(M))^{\leq B}.$$

Given a node $v$, the operator returns

$$\mathrm{Retrieve}^M(v) = \big(p(v), \ \mathrm{Out}(v)\big),$$

where $p(v) \in \mathcal{P}$ is the payload and $\mathrm{Out}(v)$ is a list of at most $B$ outgoing edges $(r, v')$, where $r$ is the edge's relation and $v'$ is its target node. This abstraction matches common KG implementations where a "node fetch" returns attributes/text plus a bounded set of neighbor references.

## C.2. Population risk and approximation error

**Definition C.1** (Population risk). For a predictor $\pi$ and distribution $D$, define the population risk under 0–1 loss:

$$L(\pi; D) := \mathbb{E}_{(M,x,y) \sim D}\big[\mathbf{1}[\pi(x, M) \neq y]\big].$$

**Definition C.2** (Approximation error). For a hypothesis class $\mathcal{H}$ and distribution $D$, define

$$\mathrm{opt}(\mathcal{H}; D) := \inf_{\pi \in \mathcal{H}} L(\pi; D).$$

A policy that learns nothing about $y$ can still guess using the prior. To capture this baseline under 0–1 loss, define

$$\varepsilon_Y := 1 - \sup_{y \in \mathcal{Y}} P_Y(y).$$

Intuitively, $\varepsilon_Y$ is the minimum error achievable when you only know the prior $P_Y$ and have no additional information about $y$.

## C.3. Active vs. passive retrieval

We now define active and passive policies, given an LM with parameter $\theta \in \Theta$ and a retrieval budget of $T$ steps. The LM induces functions $Q_{\theta,t}$ for $t \leq T$ which decide which node to retrieve in step $t$, and a final answer head $H_\theta$ which outputs an answer given the retrieval history.

Under the active policy, $Q_{\theta,t}$ is conditioned on the entire history seen so far, but under the passive policy, it is conditioned only on the query. Let $v^{(t)}$ denote the node the LM chooses to retrieve in step $t$. Formally:

**Definition C.3** (Active LM+retrieval policy class). An *active* policy with parameters $\theta$ proceeds as:

$$v^{(1)} = Q_{\theta,1}(x), \qquad z^{(1)} = \mathrm{Retrieve}^M(v^{(1)}),$$

and for $t = 2, \ldots, T$,

$$v^{(t)} = Q_{\theta,t}(x, z^{(1)}, \ldots, z^{(t-1)}), \qquad z^{(t)} = \mathrm{Retrieve}^M(v^{(t)}).$$

The prediction is

$$\pi_\theta^{\mathrm{act}}(x, M) = H_\theta(x, z^{(1)}, \ldots, z^{(T)}).$$

Define the hypothesis class

$$\mathcal{H}_{\mathrm{active}}^{\mathrm{LM}}(T) := \{\pi_\theta^{\mathrm{act}} : \theta \in \Theta\}.$$

**Definition C.4** (Passive LM+retrieval policy class). A *passive* policy with parameters $\theta$ proceeds as:

$$v^{(t)} = Q_{\theta,t}^{\text{pass}}(x), \qquad t = 1, \ldots, T,$$

then retrieves $z^{(t)} = \text{Retrieve}^M(v^{(t)})$ for all $t$, and outputs

$$\pi_\theta^{\text{pass}}(x, M) = H_\theta(x, z^{(1)}, \ldots, z^{(T)}).$$

Define

$$\mathcal{H}_{\text{passive}}^{\text{LM}}(T) := \{\pi_\theta^{\text{pass}} : \theta \in \Theta\}.$$

## C.4. Main theorem: active retrieval is strictly more powerful than passive

We show that allowing the retriever to adapt its queries to previously retrieved content yields a strictly larger class of predictors than any non-adaptive (query-only) retriever.

**Theorem C.5** (Active retrieval is strictly more powerful than passive). *For any retrieval budget $T \geq 2$, the passive hypothesis class is strictly contained in the active hypothesis class:*

$$\mathcal{H}_{\text{passive}}^{\text{LM}}(T) \subsetneq \mathcal{H}_{\text{active}}^{\text{LM}}(T).$$

To start, we first observe that passive policies are a subset of active policies, making the active hypothesis class *at least* as strong as the passive class.

**Lemma C.6.** *For any retrieval budget $T$, we have*

$$\mathcal{H}_{\text{passive}}^{\text{LM}}(T) \subseteq \mathcal{H}_{\text{active}}^{\text{LM}}(T).$$

*Proof.* Fix $T$ and any passive policy $\pi_\theta^{\text{pass}} \in \mathcal{H}_{\text{passive}}^{\text{LM}}(T)$. By definition, its retrieval queries take the form $v^{(t)} = Q_{\theta,t}^{\text{pass}}(x)$, depending only on the query $x$. Define an active policy $\tilde{\pi}_\theta^{\text{act}}$ with queries

$$\tilde{v}^{(t)} = \tilde{Q}_{\theta,t}(x, z^{(1)}, \ldots, z^{(t-1)}) := Q_{\theta,t}^{\text{pass}}(x), \qquad t = 1, \ldots, T,$$

i.e., the active policy ignores the retrieval history and issues the same sequence of nodes as the passive one. Using the same answer head $H_\theta$, the resulting prediction function satisfies $\tilde{\pi}_\theta^{\text{act}}(x, M) = \pi_\theta^{\text{pass}}(x, M)$ for all $(x, M)$. $\square$

**What remains to be shown.** To prove strictness in Theorem 1 for any $T \geq 2$, it now suffices to construct a distribution $D$ (depending on $T$) such that active retrieval can achieve zero error with budget $T$, while any passive policy with the same budget has error bounded away from zero, which we will show in the next subsection.

## C.5. A separating task family: Binary-Tree Needle-in-a-Haystack

We now construct a task with the required property. Intuitively, the query identifies a root node. Retrieval from the root reveals two candidate children; the root payload contains a single bit telling which child to follow. Repeating for $d$ hops yields a unique target leaf, whose payload contains the answer label. Active retrieval can follow the revealed bits in order to achieve zero error, but passive retrieval must guess the leaf, incurring irreducible error.

**Binary tree.** We form a complete binary tree of depth $d = T - 1$. The root is denoted $v_\emptyset$. Each internal node $v_u$ is indexed by a binary string $u$ representing the path from the root to this node: starting from the root, each bit indicates whether to take the left (0) or right (1) child. As such, each internal node $v_u$ has two children, labeled $v_{u0}$ and $v_{u1}$.

**Ground-truth leaf.** Sample a target leaf index $u^\star \in \{0,1\}^d$ uniformly at random, and let the target node be the corresponding leaf $v_{u^\star}$. We interpret the bits of $u^\star = (u_1^\star, \ldots, u_d^\star)$ as the ground-truth path from the root: starting at $v_\emptyset$, at depth $t$ we follow the child indexed by $u_t^\star \in \{0,1\}$, so the unique root-to-leaf path is

$$v_\emptyset \rightarrow v_{u_1^\star} \rightarrow v_{u_1^\star u_2^\star} \rightarrow \cdots \rightarrow v_{u_1^\star \cdots u_d^\star} = v_{u^\star}.$$

To make this path discoverable by active retrieval, we encode the next bit along the target path in the payloads of nodes on the path: for each prefix $u = u_1^\star \cdots u_t^\star$ with $|u| = t < d$,

$$p(v_u) \text{ encodes the next bit } u_{t+1}^\star.$$

**Answer label.** Sample $y \sim P_Y$. The answer label is encoded *only* in the target leaf payload:

$$p(v_{u^\star}) \text{ encodes } y.$$

All other returned information (payloads of non-target nodes and outgoing lists) is independent of $y$. In particular, unless a policy retrieves $v_{u^\star}$, it obtains no information about $y$ beyond the prior.

**Query.** The query $x$ identifies the root node $v_\emptyset$ and asks for the label $y$.

**Definition C.7** (Binary-Tree Needle-in-a-Haystack distribution $D_{n,d}$). Fix $d \geq 1$ and set $n$ as:

$$n \;=\; 1 + 2 + \cdots + 2^d \;=\; 2^{d+1} - 1.$$

The distribution $D_{n,d}$ over triples $(M, x, y) \in \mathcal{M}_n \times \mathcal{X}_n \times \mathcal{Y}$ is defined as follows:

1. Sample a target leaf index $u^\star \sim \mathrm{Unif}(\{0,1\}^d)$.

2. Sample $y \sim P_Y$.

3. Construct a heterogeneous KG memory $M \in \mathcal{M}_n$ as the binary tree above:
   - for each prefix $u = u_1^\star \cdots u_t^\star$ with $t < d$, the payload $p(v_u)$ encodes the next bit $u_{t+1}^\star$;
   - the payload $p(v_{u^\star})$ encodes $y$;
   - all other node payloads are independent of $y$.

4. Output a query $x$ that identifies $v_\emptyset$ and asks for the label stored at $v_{u^\star}$.

### C.6. Active retrieval achieves zero error

**Lemma C.8.** *For the distribution $D_{n,d}$, there exists a parameter setting $\theta$ and a budget $T = d + 1$ such that*

$$L\big(\pi_\theta^{\mathrm{act}}; D_{n,d}\big) = 0.$$

*Consequently,*

$$\mathrm{opt}\big(\mathcal{H}_{\mathrm{active}}^{\mathrm{LM}}(d+1); D_{n,d}\big) = 0.$$

*Proof.* Consider the following active strategy. The query $x$ identifies the root node $v_\emptyset$. Initialize $u := \emptyset$. For $t = 0, 1, \ldots, d-1$:

1. Retrieve the current node $v_u$ to obtain $\mathrm{Retrieve}^M(v_u) = (p(v_u), \mathrm{Out}(v_u))$. From $p(v_u)$, read the next-bit value $b \in \{0,1\}$ (which equals $u_{t+1}^\star$ when $u$ is the length-$t$ prefix of $u^\star$).

2. Move to the corresponding child: set $u \leftarrow ub$, i.e., go to $v_{u0}$ if $b = 0$ and to $v_{u1}$ if $b = 1$.

After $d$ steps we have reached the target leaf $v_{u^\star}$. Finally retrieve $v_{u^\star}$ to read its payload, and output $y$. This uses at most $d + 1$ retrievals and is correct for every sample $(M, x, y) \sim D_{n,d}$. $\square$

### C.7. Passive retrieval has irreducible error unless the budget is exponential

**Lemma C.9.** *For any passive policy with budget $T$, uniformly over all parameters $\theta$,*

$$L(\pi_\theta^{\mathrm{pass}}; D_{n,d}) \;\geq\; \varepsilon_Y \left(1 - \frac{T}{2^d}\right), \qquad \text{where} \qquad \varepsilon_Y := 1 - \sup_{y \in \mathcal{Y}} P_Y(y).$$

*Proof.* Fix $\theta$. A passive policy chooses all nodes $v^{(1)}, \ldots, v^{(T)}$ to retrieve as functions of $x$ alone, before observing any retrieved values.

Under $D_{n,d}$, the target leaf index $u^\star \in \{0,1\}^d$ is uniform and is not revealed by $x$. Let $S := \{v^{(1)}, \ldots, v^{(T)}\}$ be the (multi)set of retrieved nodes, and let $L_d := \{v_u : u \in \{0,1\}^d\}$ be the set of leaves. Since only leaves can equal the target $v_{u^\star}$,

$$\Pr[v_{u^\star} \in S] = \Pr[v_{u^\star} \in S \cap L_d] = \frac{|S \cap L_d|}{2^d} \leq \frac{T}{2^d}.$$

The label $y$ is encoded only in the target payload $p(v_{u^\star})$. Therefore, the policy can obtain information about $y$ only if it retrieves $v_{u^\star}$. Otherwise, the entire retrieval transcript is independent of $y$. In that case, the best possible prediction depends only on the prior $P_Y$, and must incur error at least $\varepsilon_Y$. Therefore,

$$L(\pi_\theta^{\text{pass}}; D_{n,d}) \geq \Pr[\text{miss target}] \cdot \varepsilon_Y \geq \left(1 - \frac{T}{2^d}\right) \varepsilon_Y.$$

$\square$

### C.8. Strictness and approximation-theoretic separation

*Proof of Theorem 1.* Fix any $T \geq 2$ and set $d := T - 1$. The inclusion $\mathcal{H}_{\text{passive}}^{\text{LM}}(T) \subseteq \mathcal{H}_{\text{active}}^{\text{LM}}(T)$ follows by taking any passive policy and viewing it as an active policy whose query functions ignore the history and depend only on $x$ (Lemma C.6).

For strictness, consider the separating distribution $D_{n,d}$ from Definition 5. By Lemma C.8, $\text{opt}(\mathcal{H}_{\text{active}}^{\text{LM}}(T); D_{n,d}) = 0$ (since $T = d + 1$). By Lemma C.9,

$$\text{opt}(\mathcal{H}_{\text{passive}}^{\text{LM}}(T); D_{n,d}) \geq \varepsilon_Y \left(1 - \frac{T}{2^d}\right) = \varepsilon_Y \left(1 - \frac{T}{2^{T-1}}\right).$$

If the two hypothesis classes were equal, they would have equal $\text{opt}(\cdot; D)$ for every distribution $D$, contradicting the strict gap above. Hence the inclusion is strict for every $T \geq 2$. $\square$

## D. Experiments

### D.1. Datasets

**LoCoMo** (Maharana et al., 2024). LoCoMo is a benchmark designed to evaluate long-term conversational memory in extended dialogue settings. The dataset consists of 50 conversations generated via a human–LLM pipeline. Each conversation spans up to 35 sessions, with an average length of approximately 300 turns, and is accompanied by roughly 200 question–answer pairs. LoCoMo includes multiple question categories, covering single-hop, multi-hop, temporal, and open-domain queries, which require retrieving and integrating information from distant parts of the conversation history. In our experiments, we exclude adversarial questions, as most baseline methods do not support this setting and the task primarily evaluates the ability to detect unanswerable queries rather than memory reconstruction or multi-hop reasoning.

**LongMemEval** (Wu et al., 2025). LongMemEval is a benchmark designed to evaluate very long-term memory capabilities of chat assistants under sustained user–assistant interactions. Each evaluation instance consists of a sequence of timestamped chat sessions, followed by a question that requires recalling and reasoning over the interaction history. We adopt the LongMemEval-S setting, which contains approximately 500 questions, each paired with a chat history of around 115K tokens. This setting provides a challenging evaluation of long-term memory. In our experiments, we primarily focus on the single-session-user multi-session, temporal-reasoning and single-session-preference question types.

### D.2. Baselines

**Retrieval-Augmented Generation.** Retrieval-augmented generation(RAG) stores dialogue histories as text chunks in a vector database. For each query, the system retrieves the top-$k$ most similar memory entries based on embedding similarity and concatenates them with the query to form the input context.

**A-Mem** (Xu et al., 2025). A-Mem constructs structured memory notes for each conversational episode, where new memory notes are linked to existing ones using LLM-assisted analysis, resulting in a directed graph structure. For retrieval, A-MEM embeds the query to retrieve relevant memory notes and then expands to their neighboring nodes.

**MemoryOS** (Kang et al., 2025). MemoryOS organizes agent memory into a three-tier hierarchy, including short-term memory for recent dialogue context, mid-term memory for topic-based interaction history, and long-term persona memory for stable user and agent characteristics. For retrieval, it performs hierarchical memory access across all layers and the retrieved memories are concatenated with persona information to form the input context.

**LangMem** (LangChain, 2025). LangMem stores conversational memories by embedding each dialogue turn and maintaining them in a vector database via a memory management tool. At inference time, it retrieves the top-$k$ most relevant memories based on embedding similarity and summarizes them using the LLM to support response generation.

**Mem0** (Chhikara et al., 2025). Mem0 incrementally extracts salient natural-language facts from conversations and maintains a compact long-term memory through LLM-driven update operations. For retrieval, it performs similarity-based search over stored memories and incorporates the retrieved facts into the input context for answer generation.

### D.3. Evaluation Metric

**LLM-Judge.** We adopt an LLM-based judge to evaluate the semantic correctness of generated answers. Given a model-generated answer $\hat{y}$ and a reference answer $y$, the judge determines whether $\hat{y}$ is semantically equivalent to $y$, allowing for paraphrasing and differences in surface form. The judge outputs a binary decision indicating correctness.

**$F_1$ Score.** In addition to Recall, we report the $F_1$ score computed from the judge decisions. For each question, the generated answer is treated as a positive prediction, and the judge decision determines whether it is a true positive or a false positive. Aggregating over the dataset, precision and recall are computed as:

$$\text{Precision} = \frac{\sum_i J(\hat{y}_i, y_i)}{\sum_i \mathbb{I}[\hat{y}_i \neq \emptyset]}, \qquad \text{Recall} = \frac{\sum_i J(\hat{y}_i, y_i)}{N}, \tag{22}$$

and the $F_1$ score is defined as:

$$F_1 = \frac{2 \cdot \text{Precision} \cdot \text{Recall}}{\text{Precision} + \text{Recall}}. \tag{23}$$

This formulation penalizes both incorrect answers and failures to produce an answer, and is particularly suitable for open-ended question answering settings.

**Evidence Recall (Recall).** Evidence Recall measures the fraction of ground-truth supporting evidence that is successfully retrieved during memory reconstruction. For each question $i$, let $\mathcal{E}_i^*$ denote the set of annotated ground-truth evidence items, and let $\hat{\mathcal{E}}_i$ denote the set of evidence items retrieved by the agent. Evidence Recall is computed as:

$$\text{Recall} = \frac{1}{N} \sum_{i=1}^{N} \frac{|\hat{\mathcal{E}}_i \cap \mathcal{E}_i^*|}{|\mathcal{E}_i^*|}, \tag{24}$$

where $N$ is the total number of evaluation questions. This metric reflects the effectiveness of the retrieval process in recovering relevant supporting evidence, independent of final answer generation.

### D.4. Implementation

We use GPT-4o-mini as the LLM judge with temperature set to 0.0. Each method is evaluated three independent times, and we report the mean and standard deviation of the judge scores across runs. To ensure comparable compute budgets across methods, we cap the agent's reasoning to at most 8 turns per query, and allow up to 10 tool invocations per turn. Agents may terminate early if a stopping condition is met before exhausting the budget.

### D.5. Detailed Results on LONGMEMEVAL

Table 5 presents detailed results on LONGMEMEVAL under different evaluation settings, evaluated by $F_1$ and LLM-Judge.

### D.6. Budget Sensitivity Analysis of Multi-step Reconstruction

Figure 9 analyzes the trade-off between the number of reasoning turns and per-turn parallel retrieval on multi-hop questions in LOCOMO. We vary the per-turn retrieval budget ($K$), corresponding to the maximum number of parallel tool calls allowed within a single reasoning turn, and the maximum number of reasoning turns ($T$), while keeping other settings fixed.

*Table 5.* Performance on LONGMEMEVAL evaluated by $F_1$ and LLM-Judge $\uparrow$. Best and second-best results among Gemini-backbone methods are marked in **bold** and underline, respectively. MRAgent* uses Claude for retrieval while memories are constructed by Gemini.

| Method | multi-session | | single-user | | temporal-reasoning | | single-preference | | Overall |
| --- | --- | --- | --- | --- | --- | --- | --- | --- | --- |
| | $F_1 \uparrow$ | $J \uparrow$ | $F_1 \uparrow$ | $J \uparrow$ | $F_1 \uparrow$ | $J \uparrow$ | $F_1 \uparrow$ | $J \uparrow$ | $J \uparrow$ |
| RAG | 45.00 | 54.89 | 77.94 | 85.71 | 43.88 | 42.86 | 5.42 | 33.33 | 54.65 |
| A-Mem | 28.26 | 42.85 | 75.30 | 90.00 | 34.18 | 45.11 | 9.09 | 46.43 | 52.98 |
| MemoryOS | 45.02 | 56.39 | 79.50 | 87.14 | 35.67 | 38.35 | 9.53 | 46.67 | 54.92 |
| LangMem | 41.14 | 52.63 | 72.43 | 78.57 | 36.80 | 45.71 | 6.32 | 36.67 | 53.77 |
| Mem0 | 37.53 | 50.38 | 72.43 | 78.57 | 35.64 | 45.11 | 5.89 | 40.00 | 53.01 |
| MRAgent | **49.92** | **68.42** | **80.99** | **92.85** | **50.16** | **68.42** | **23.96** | **66.67** | **72.95** |
| MRAgent* | 66.31 | 86.46 | 82.41 | 92.85 | 60.10 | 85.71 | 15.58 | 78.57 | 86.76 |

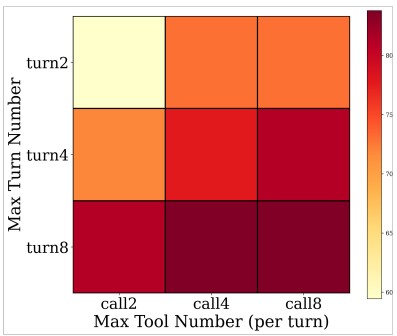

*Figure 9.* Performance on multi-hop queries in LOCOMO as a function of the number of reasoning turns ($T$) and the per-round retrieval budget ($K$), evaluated under the Claude backbone using LLM-Judge (J).

**Reconstruction depth cannot be substituted by increased parallel exploration.** As the number of reasoning turns $T$ increases, accuracy improves steadily and monotonically across all values of $K$, with deeper reconstruction yielding substantially higher performance. In contrast, increasing the per-turn retrieval budget $K$ leads to only limited gains that quickly saturate. These results indicate that while parallel exploration increases retrieval breadth within a single reasoning turn, it cannot replace the sequential composition of evidence enabled by multi-turn reconstruction.

### D.7. Evidence Coverage by Retrieval Operators

To analyze how different retrieval operators contribute to memory reconstruction, we examine the evidence coverage of each tool on LOCOMO, aggregated by question category. Table 6 reports the coverage rates of individual operators, reflecting their functional roles during reconstruction.

**Different retrieval operators specialize in distinct query structures.** Temporal questions are predominantly resolved through `query_conversation_time`, which accounts for the majority of temporally grounded evidence. Multi-hop questions rely heavily on `query_tag_events` and `query_topic_events`, indicating that associative expansion over tags and topics is essential for recovering evidence distributed across multiple episodes. In contrast, open-domain questions exhibit more balanced coverage across multiple operators, reflecting the need to integrate episodic, semantic, and contextual information. Overall, these results demonstrate that MRAgent performs structured, operator-dependent memory reconstruction. Rather than relying on uniform or similarity-driven retrieval, the agent selectively activates different operators based on query structure, resulting in differentiated evidence acquisition patterns.

### D.8. Case Studies

As shown in Figure 7, we present a case study illustrating how MRAgent performs multi-turn memory reconstruction over a structured memory graph. The query, "Which of Joanna's screenplays were rejected by production companies?", requires linking screenplay submission events with subsequent rejection events that are distributed across multiple dialogue sessions.

**MRAgent incrementally reconstructs relevant evidence through iterative graph exploration and multi-turn reasoning.** In the first reasoning turn, the agent retrieves screenplay submission and rejection events by traversing tag-based associations,

*Table 6.* Evidence coverage of different tools across question types on LOCOMO under the Claude backbone.

| Tool | Multi-hop | Temporal | Open domain | Single-hop |
|------|-----------|----------|-------------|------------|
| `query_tag_events` | 66.33 | 81.08 | 41.18 | 74.76 |
| `query_conversation_time` | 4.08 | 86.49 | 17.65 | 3.88 |
| `query_event_context` | 18.37 | 32.43 | 35.29 | 33.01 |
| `query_personal_aspect` | 21.43 | 2.70 | 29.41 | 5.83 |
| `query_topic_events` | 33.67 | 45.95 | 35.29 | 27.18 |

identifying candidate events that are directly related to the query. In the second turn, it expands these candidates by querying event-level context and keywords to obtain detailed information about each rejection. In the third and fourth turns, the agent queries semantic information about Joanna to recover properties of her screenplays, enabling a clearer characterization of each submission. Finally, in the fifth turn, the agent queries temporal information to align screenplay submissions with corresponding rejection events and verify their ordering.

After five reasoning steps, MRAgent correctly infers that Joanna's first and third screenplays were rejected. This example demonstrates that answering complex, multi-session queries benefits from active, multi-step memory reconstruction that integrates associative expansion with semantic and temporal verification.

# E. Prompt

**MRAgent QA and Tool-Use Prompt**

You are a question-answering agent with access to event-based memory. Answer the question if sufficient evidence exists; otherwise, query memory tools to gather more information.

**Answer Rules:**

- Yes/No questions: output `Yes`, `No`, `Likely yes`, or `Likely no`.

- Location questions: answer with a specific place name.

- Counting questions: answer with the number of relevant items.

- Other questions: output the minimal concrete entity or phrase.

**Choose ONE mode:**

**(1) `"answer"`** — if evidence is sufficient, output:

```
{
  "mode": "answer",
  "answer": "...",
  "supports": ["D1:1","D1:2"],
  "confidence": 0.0-1.0
}
```

**(2) `"navigate"`** — if evidence is insufficient, call all relevant memory tools directly (no explanations).

**Keyword Extraction Prompt**

You are an information extraction system. **Only output valid JSON.**

**Task:** For each input sentence, extract **2–30 keywords** *directly from the original text*.

- Do not invent, paraphrase, or generalize keywords.

- Only include words or phrases that explicitly appear in the text.

- Do not include inferred or implicit concepts.

- Extract all explicit keywords of the following types if present: `entity`, `topic`, `verb`, `time`, `location`, `task`, `event`, `people`.

**ID Rule:** `sentence_id` must exactly match the `id` field in `TEXT`. Do not create or modify IDs.
**Output Schema (single-line JSON):**

```
{
  "sentence": [
    {
      "sentence_id": "D1:1-1",
      "keyword": ["Coraline", "park"]
    }
  ]
}
```

### Dialogue Processing Prompt

You are a dialogue processor. **Only output valid JSON.**
**Task:** For each sentence in the dialogue:

- Preserve every original sentence.

- Replace all pronouns with explicit entities or noun phrases from context.

- Do not modify verbs, adjectives, or other words.

- Assign a short concrete `tag` (at most two words).

- Normalize time to `YYYY-MM-DD` using `conversation_time`.

- If a question is answered by the next sentence, merge them.

**Topics:** Derive at least ten concrete topics overall. Assign topic IDs (`t1..tn`); each sentence lists applicable topics or `[]`.
**Personal Information:** Extract person-related facts into `personal_sentences`. If a fact appears in a sentence, also add a concise normalized version.
**ID Rules:** `id` must be `origin:number`, where `origin` exactly matches `dia_id`. Do not invent new IDs.
**Output Schema (single-line JSON):**

```
{
  "conversation_time": "YYYY-MM-DD",
  "sentence": [
    {
      "id": "D1:1-1",
      "text": "sentence.",
      "tag": "short tag",
      "origin": "D1:1",
      "topic": ["t1","t3"],
      "time": "YYYY-MM-DD"
    }
  ],
  "topics": {
    "t1": "Topic description",
    "t2": "Topic description"
  },
  "personal_sentences": [
    {
      "id": "p1",
      "text": "Normalized personal fact.",
      "tag": "preference",
      "origin": "D1:1",
      "person": "Name"
    }
  ]
}
```

