# OpenReview forum: "Memory is Reconstructed, Not Retrieved:  Graph Memory for LLM Agents"
_ICML.cc/2026/Conference — ICML 2026 regular_

### Official Review · Reviewer_tm5v · 2026-03-09

**Soundness:** 3
**Presentation:** 3
**Significance:** 3
**Originality:** 3
**Overall Recommendation:** 4
**Confidence:** 4

**Summary:**

This paper addresses the challenge of enabling LLM agents to reason over long interaction histories, which is difficult due to limited context windows and the limitations of current external memory mechanisms. Existing approaches typically follow a static “retrieve-then-reason” paradigm, where relevant memories are retrieved once (e.g., via similarity search) and then passed to the language model for reasoning. The authors argue that this pipeline is rigid and prevents the agent from adapting memory access as new intermediate reasoning evidence emerges.

To overcome this limitation, the paper proposes MRAgent, a framework that treats memory retrieval as an active reconstruction process rather than passive retrieval. The system represents memory as a Cue–Tag–Content graph, where Cues correspond to observable signals from the current context and tags serve as semantic connectors between related pieces of information, and
Content stores the actual memory items.

Using this associative memory graph, the agent can iteratively explore and prune memory paths during reasoning. Instead of retrieving a fixed set of documents at the start, the LLM progressively reconstructs relevant memories by following graph connections and updating the retrieval process based on intermediate reasoning results. This design aims to make memory access more adaptive while avoiding combinatorial explosion during exploration.

The authors evaluate the proposed approach on long-horizon memory reasoning benchmarks, including LOCO and LongMemEval. Experimental results show that MRAgent improves task performance compared to strong baselines while also reducing token usage and runtime cost, demonstrating the efficiency of active memory reconstruction.

**Compliance With Llm Reviewing Policy:**

Affirmed.

**Final Justification:**

Even after the detailed rebuttal and additional experiments, some of my concerns still remain partially unresolved. In particular, questions around the extent of dependence on the underlying LLM and robustness across settings are not fully convincing.

That said, the authors have provided substantial clarifications and empirical support, which strengthen the paper overall. I believe the work still meets the bar for acceptance, so I am keeping my original score.

**Key Questions For Authors:**

Q1) The proposed system relies heavily on the reasoning capabilities of the underlying LLM for several key components, including cue generation, tag creation, and traversal decisions in the memory graph. Have the authors evaluated how sensitive the performance of the system is to the choice of LLM (e.g., comparing stronger vs. smaller models)?

Q2)The Cue-Tag content memory graph relies on LLM-based extraction of cues and tags during memory construction. How robust is the system to errors in this stage? For example, if cues or tags are incorrectly generated, does the active reconstruction process still recover the relevant information?

Q3)Active reconstruction involves multiple reasoning and traversal steps over the memory graph. How does the computational cost (e.g., number of LLM calls or latency) scale with the size of the memory graph or length of the interaction history?

**Limitations:**

The authors include a discussion of the limitations of their approach and potential risks associated with deploying LLM-based memory systems. They acknowledge issues such as dependence on the underlying language model, possible inaccuracies in memory reconstruction, and challenges related to scalability and reliability. This discussion demonstrates appropriate awareness of the system’s constraints and responsible consideration of its potential impacts.

**Strengths And Weaknesses:**

1. Soundness

Strengths

1. The paper proposes a technically coherent framework for active memory reconstruction in LLM agents, integrating reasoning and memory access into a unified process.
2. The core idea allowing the LLM to iteratively explore a structured memory graph rather than relying on one-shot retrieval is well-motivated and clearly formalized.
3. The proposed Cue–Tag–Content (CTC) memory graph provides a structured representation that separates cues, associative tags, and memory contents, enabling semantically guided retrieval.
4. This design helps control graph traversal and reduces combinatorial expansion when exploring memory nodes.
5. The reconstruction procedure is explicitly defined with an iterative state update process, including the active set of candidate nodes and accumulated evidence, which allows adaptive memory exploration during reasoning.
6. The empirical evaluation is reasonably comprehensive. The method is compared against several representative memory systems (e.g., RAG-based approaches and structured memory frameworks) across two long-context benchmarks, LOCOMO and LongMemEval, demonstrating consistent improvements in answer quality.
7.The paper also provides ablation studies isolating the contributions of the memory structure and multi-step reasoning mechanism, showing that both contribute meaningfully to the final performance gains.
A theoretical argument is included suggesting that adaptive retrieval policies are strictly more expressive than passive retrieval policies, providing conceptual support for the proposed paradigm.


Weaknesses

1. The theoretical analysis is largely conceptual and abstraction-level, and the practical implications of the expressivity result are not empirically validated. It is unclear how strongly the theoretical advantage translates to real-world agent settings.
2. While the experiments demonstrate improvements, they rely on LLM-as-judge evaluation and a limited number of benchmarks, which may introduce evaluation noise and limit generalizability.
3. The memory construction pipeline depends heavily on LLM-based extraction of cues and tags, which may introduce errors or instability. The paper does not extensively analyze how sensitive the system is to these upstream extraction errors.
4. Some implementation choices (e.g., traversal budgets, tool invocation limits, reasoning steps) appear somewhat heuristic, and the paper provides limited discussion about how robust the method is to these hyperparameters.

3. Significance

Strengths

1. The paper addresses a central challenge in LLM agents: reasoning over long interaction histories with limited context windows.
2.Improving long-horizon memory systems is important for applications such as conversational assistants, multi-session agents, and decision-support systems, making the problem practically relevant.
4. The idea of integrating reasoning directly into retrieval could influence future work on retrieval-augmented systems, agent architectures, and memory-based reasoning.
5.The reported improvements on benchmarks like LOCOMO and LongMemEval suggest the method can improve both accuracy and computational efficiency, which is valuable for real-world deployment.

---

> ### Author Rebuttal · Authors · 2026-03-30
>
> We thank the reviewer for the detailed feedback and for recognizing the technical coherence of our framework, its potential influence on future retrieval-augmented systems, and the comprehensive experimental validation.
>
> **W1: (Theoretical analysis)** **The theorem establishes an exponential quantitative gap:** passive retrieval incurs irreducible error at least $ε_Y(1−T/2^d)$ on our constructed task family, while active retrieval achieves zero error with budget T = d+1. Matching this performance passively would require a budget on the order of 2^d. This directly motivates MRAgent’s choice of an **iterative multi-round reconstruction** architecture over **single-round large-scale parallel retrieval**(existing methods). This quantitative result also directly explains our experimental finding that increasing reconstruction depth yields monotonic gains while increasing parallel retrieval breadth saturates quickly (Figure 9) .
>
> **The theorem directly motivated concrete design choices:** (1) *The necessity of intermediate navigational structures.* Internal nodes in the binary-tree construction encode navigational guidance rather than final answers. This motivates the Tag layer as lightweight navigational signals that allow the LLM to assess retrieval directions before committing to expensive episodic content.   (2) *LLM-guided adaptive retrieval, not merely multi-step retrieval.* The theorem shows that the gain comes from conditioning each retrieval decision on accumulated evidence, which motivates our three-stage loop (LLM Reasoning → Controlled Traversal → LLM Routing), where the LLM actively controls retrieval direction.
>
> **W2: (LLM Judge)** We follow **standard evaluation protocols used in prior work on these benchmarks**, ensuring fair and consistent comparison with existing methods. In addition, to mitigate noise: (1) we repeat experiments multiple times — as shown in Table 1,  standard deviations are below 1.0 across all settings,  indicating low evaluation variance;  (2) we report both LLM-judge and F1 metrics (Table 1, Table 5), showing consistent trends; (3) we evaluate on multiple benchmarks (LoCoMo and LongMemEval) covering different types of long-horizon memory tasks. LoCoMo and LongMemEval are currently the two most widely used benchmarks for long-term conversational memory, covering different settings (single-conversation, multi-session) and question types (single-hop, multi-hop, temporal, open-domain, preference).
>
> **W3/Q2: (Extraction Sensitivity)**  The system exhibits meaningful robustness to upstream extraction quality due to its reconstruction process.  (1) The CTC structure provides redundant associative connectivity — the same content is typically reachable through multiple cues and tags, so individual extraction errors do not block access. (2) The reasoning mechanism allows the agent to discover alternative cues from intermediate evidence and redirect traversal, partially self-correcting upstream errors. (3) Unlike fixed similarity matching, LLM-guided traversal can infer implicit semantic relations even when surface-level cues are imprecise.
>
> To disentangle the effects of memory construction quality and retrieval capability, we conduct controlled experiments varying the LLM used for each stage:
>
> |Construction LLM|Retrieval LLM|LoCoMo J ↑ |
> |--|--|--|
> |Gemini-2.0-Flash|Gemini-2.0-Flash|63.55|
> |Gemini-2.0-Flash|Gemini-2.5-Flash|79.63|
> |Gemini-2.5-Flash|Gemini-2.5-Flash|84.21|
>
> This result highlights the dominant role of adaptive reconstruction: when we fix memory construction and only strengthen the retrieval-stage model, performance improves by +16.1, demonstrating that the gain primarily comes from adaptive reasoning rather than memory quality.
>
> **W4/Q3: (Computational Cost)** We analyze hyperparameter sensitivity in Appendix D.6 . The number of LLM calls scales with reasoning turns (T) and per-turn tool budget (K), not with graph size. Figure 6(b) shows most queries converge within 2–3 turns, with K capped at 10. Graph traversal operations (tag lookup, content retrieval) are O(1) lookups. Total cost per query is bounded by O(T × K). The cost depends on reasoning depth rather than memory size, making the approach naturally scalable to long-horizon settings. On LongMemEval, MRAgent achieves 586s per sample while the strongest-performing baseline (MemoryOS) requires 3,136s — a 5× speedup with substantially better accuracy (72.95 vs. 54.92 J score). This confirms that active reconstruction not only improves quality but also scales efficiently to larger graphs.
>
> **Q1: (LLM Model)** Tables 1–2 in paper show consistent improvements across both Gemini-2.5-Flash and Claude-Sonnet-4.5.  We additionally evaluate a weaker backbone Gemini-2.0-Flash:
>
> |Backbone(Construction + Retrieval)| LoCoMo J ↑ | Δ vs. best baseline|
> |--|--|--|
> |Gemini-2.0-Flash|63.55|+13.1%|
> |Gemini-2.5-Flash|84.21|+23.3%|
> |Claude-Sonnet-4.5 |88.32 |+12.4%|
>
> The relative improvement holds even with the weaker Gemini-2.0-Flash.

---

> > ### Author Rebuttal · Reviewer_tm5v · 2026-04-03
> >
> > The rebuttal has addressed several of my questions satisfactorily. In particular, the authors clarified that the theoretical motivation behind active reconstruction and how it informs the design choices ,they have given dditional details on evaluation robustness (e.g., variance, multiple metrics), done scalability analysis showing that cost depends on reasoning depth rather than memory size.Additional experiments demonstrating consistent gains across different LLM backbones.
> >
> > But still some concerns remain in the paper
> >
> > 1)While the authors show that improvements hold across different models, the system still heavily relies on the LLM for cue/tag extraction,traversal decisions and reasoning control.
> > It remains somewhat unclear whether the gains come primarily from the proposed memory framework or from the strength of the underlying model. In particular, the efficiency and robustness of the system under weaker or less capable models is still not fully convincing.
> >
> > 2) The rebuttal argues that redundancy in the graph mitigates extraction errors, but this is not quantitatively evaluated. It would be helpful to see more controlled experiments analyzing how performance degrades under noisy or incorrect cue/tag generation.
> >
> > 3)Although the asymptotic cost is bounded by 𝑂(𝑇×𝐾) O(T×K), the multi-step reasoning loop introduces additional LLM calls and system complexity. It remains somewhat unclear whether the practical efficiency gains consistently hold across different settings, especially in resource-constrained scenarios.

---

> > > ### Author Response · Authors · 2026-04-06
> > >
> > > We thank the reviewer for the constructive follow-up. We address each remaining concern with new experimental evidence below.
> > >
> > > **Concern 1: LLM reliance**
> > >
> > > Thanks for the review. We note that reliance on LLM capabilities is **shared across all evaluated methods**, not unique to MRAgent. The following table summarizes each method's LLM dependence:
> > >
> > > | Method | Memory Construction (LLM function) | Memory Structure |
> > > |--|--|--|
> > > | A-Mem **(Graph Memory System)**| Note generation (**LLM: Information Extraction+Summary**) + Link generation(**LLM: Analyze potential connections with history**) | **LLM-constructed** graph |
> > > | MemoryOS | Topic extraction(**LLM: Information Summary+persona update**) + Hierarchical organization(**LLM: Relevance Evaluation**) | **LLM-organized** hierarchy |
> > > | Mem0 | Fact extraction(**LLM: Information Extraction+Summary**) + CRUD operations(**LLM: ADD/DELETE/UPDATE Operation**) | **LLM-maintained** fact store |
> > > | MRAgent | Cue/Tag extraction (**LLM: Information Extraction+Summary**) | CTC associative graph|
> > >
> > > In fact, MRAgent's **construction phase** places **less** burden on the LLM than some baselines. For example, **A-Mem** requires the LLM to analyze correlations between new content and existing memory notes and build inter-note links, **demanding strong analytical capability**. In contrast, **MRAgent** only uses the LLM to extract keywords (cues) and generate short summary phrases (tags), while the graph structure follows the original dialogue organization. Since all methods in Table Q1 use the **identical** LLM backbone, MRAgent's consistent improvements are directly attributable to the CTC associative structure and active reconstruction mechanism, not to differences in LLM capability alone.
> > >
> > > To further validate that gains stem from the framework rather than model capability,  we evaluate on Qwen3-32B:
> > >
> > > | Backbone| Mem0  | MRAgent | Δ  |
> > > | --|--|--|--|
> > > | Qwen3-32B| 59.29 | 71.34| +20.33% |
> > > | Gemini-2.5-Flash | 68.31| 84.21| +23.3% |
> > > | Claude-Sonnet| 69.02 | 88.32 | +28.0%|
> > >
> > > MRAgent consistently outperforms the strongest baseline across all three backbones. This suggests that the performance gains are largely attributable to the CTC structure and active reconstruction mechanism, not model strength.
> > >
> > > **Concern 2: Quantitative evaluation of extraction robustness**
> > >
> > > To quantify robustness to upstream extraction errors, we conduct controlled perturbation experiments on LoCoMo, randomly replacing 20% and 30% of cues and tags sampled from other episodes. We apply analogous perturbation to A-Mem's note information(nodes of graph structure). In both cases, evidence content remains unchanged.
> > >
> > > |Method|Baseline J|Corrupt 20%|Corrupt 30%|
> > > |--|--|--|--|
> > > |MRAgent|84.21 | 81.12(−3.7%) | 77.14 (−8.4%)  |
> > > |A-Mem| 55.97| 52.21(−6.7%) | 48.57 (−13.2%) |
> > >
> > > MRAgent's degradation under 30% corruption (−8.4%) is notably lower than A-Mem's (−13.2%). Moreover, MRAgent under 30% corruption (77.14) still substantially outperforms A-Mem's uncorrupted baseline (55.97).
> > >
> > > These results confirm that **tag structure is essential, but individual tag accuracy is not a bottleneck.**  Three architectural properties explain this robustness: (1) each content is linked to multiple cues/tags, so errors in individual associations do not block access; structural anchors (speaker identity, timestamps) are also invariant to extraction noise. (2) Diverse retrieval operators (semantic, contextual, temporal; Table 4, Table 6) provide independent pathways to the same evidence. (3) The underlying graph structure is built on dialogue organization. A corrupted tag lowers selection priority but does not break structural connectivity.
> > >
> > > **Concern 3: Efficiency under resource-constrained settings**
> > >
> > > To evaluate practical efficiency under limited budgets, we vary reconstruction depth (T) and per-turn tool budget (K) on the multi-session category of LongMemEval, which demands the most extensive exploration:
> > >
> > > | Config|T (max rounds) | K (max calls/round) | J ↑   | Prompt Tokens |
> > > |--| --|--|--|--|
> > > | Constrained | 2  | 8  | 57.14 | 81k  |
> > > | Moderate| 4 | 8 | 64.46 | 89k  |
> > > | Full | 8   | 8| 68.42 | 119k |
> > >
> > > | Config | T (max rounds) | K (max calls/round) | J ↑   | Prompt Tokens |
> > > | --| -- | -- | --| --|
> > > | Constrained | 8  | 2 | 62.46 |94k|
> > > | Moderate  | 8  | 4 | 66.00|103k|
> > > | Full| 8| 8| 68.42 |119k|
> > >
> > > For reference, baselines under full budget: MemoryOS achieves 56.39 with 276k tokens; A-Mem achieves 42.85 with 633k tokens. Even under the most constrained configuration (T=2), MRAgent already matches MemoryOS's full-budget performance at 3.4× fewer tokens. Under moderate budget (T=4), it surpasses all baselines.  In practice, the agent early-stops once evidence suffices, so actual costs are often below these bounds.
> > >
> > > We appreciate the reviewer’s suggestions. The additional experiments and analysis provide further evidence that the proposed framework contributes beyond model choice, while also clarifying its robustness and practical efficiency.

---

### Official Review · Reviewer_oJai · 2026-03-11

**Soundness:** 2
**Presentation:** 3
**Significance:** 2
**Originality:** 3
**Overall Recommendation:** 4
**Confidence:** 4

**Summary:**

This paper proposes MRAgent, a memory architecture for LLM agents that treats memory retrieval as an active reconstruction process rather than a fixed retrieval step. It organizes memory as a Cue–Tag–Content graph and allows the LLM to iteratively traverse the graph, selecting and pruning retrieval paths based on intermediate reasoning evidence. Experiments on two conversational memory benchmarks show that this adaptive retrieval strategy improves both reasoning accuracy.

**Compliance With Llm Reviewing Policy:**

Affirmed.

**Final Justification:**

The author's rebuttal resolved my major concerns.

**Key Questions For Authors:**

Please refer to the weakness part.

**Limitations:**

Yes.

**Strengths And Weaknesses:**

## Strengths:
1. The paper proposes a new memory structure: the Cue–Tag–Content graph provides a clear and interpretable way to organize associative relations, helping guide retrieval and reduce irrelevant expansions.

2. Strong empirical and theoretical support: The method demonstrates significant improvements on long-term memory benchmarks.

3. The paper is easy to read, and the claim is further strengthened by a theoretical analysis showing that active retrieval is strictly more expressive than passive retrieval.


## Weaknesses:

1. The paper positions prior RAG-style memory systems as mainly relying on fixed, passive retrieval. However, a large number of prior works [1-4] on adaptive or agentic RAG, search-enhanced reasoning, and deep research already allow models to actively refine queries and retrieve evidence iteratively during reasoning. The paper should discuss these approaches in the related work and refine the claim of the comparison towards prior work in the introduction.

[1] Search-o1: Agentic Search-Enhanced Large Reasoning Models

[2] Search-R1: Training LLMs to Reason and Leverage Search Engines with Reinforcement Learning

[3] OmniSearch:  Self-Adaptive Planning Agent For Multimodal RAG

[4] MC-Search: Evaluating and Enhancing Multimodal Agentic Search with Structured Long Reasoning Chains

2. Graph-based memory is not a new design choice. Organizing memory as a graph has been explored in many prior works [1-2] on LLM agent memory and graph-based memory systems. The paper would benefit from a clearer discussion of these related methods, including how the proposed Cue–Tag–Content graph differs conceptually or practically. It would also be helpful to explain why some of these graph-based approaches were not included as baselines.

[1] H-MEM: Hierarchical Memory for High-Efficiency Long-Term Reasoning in LLM Agents

[2] SGMEM: SENTENCE GRAPH MEMORY FOR LONG-TERM CONVERSATIONAL AGENTS

3. Token cost analysis may be incomplete. The paper reports reduced token consumption during the reasoning process, but another major cost likely comes from the graph construction phase, where LLMs extract cues, tags, and memory elements. It would be useful to include this construction cost in the overall token analysis and compare the total cost (construction + retrieval) with previous systems such as A-Mem.

---

> ### Author Rebuttal · Authors · 2026-03-30
>
> We thank the reviewer for the constructive feedback! We address the weaknesses as follows:
>
> **W1: (Comparison with agentic RAG)** Thanks for suggestions. We will revise the introduction and related work to discuss agentic RAG as a distinct category. However, we would like to clarify that they are fundamentally different from MRAgent. Agentic RAGs (Search-o1, Search-R1, MC-Search and OmniSearch)  are designed for **retrieving external knowledge via search engines** (Bing, Web API)  , iteratively refining queries over unstructured corpora. MRAgent operates over **structured, associative memory** connected through temporal, causal, or entity-level relations. They operate differently:
>
> **Agentic RAG - reasoning refines the query**: Query → LLM reasons → refined query → **1-shot similarity-based search** → LLM reasons → refined query → 1-shot  similarity-based search → ... (Each retrieval is **fixed similarity-based search** over the same flat index.)
>
> **MRAgent - reasoning is embedded in retrieval itself**: Query → Cues → **LLM inspects Tags** (preview directions) → **LLM selects branch** → Content → LLM discovers new Cues along graph links → LLM inspects new Tags → ... (LLM reasons on the graph structure and  actively selects the future direction.)
>
> This distinction is particularly important for memory systems, where relevant evidence is often connected through complex associations. In agentic RAG, the LLM decides *what to search for*, but retrieval itself is always embedding similarity against a flat index. No matter how many times the query is refined, if two memory items are connected through complex associations (temporal co-occurrence, causal links) , query refinement cannot discover these connections — **it is passive retrieval with a smarter query generator.** In MRAgent, the LLM decides *how to retrieve*: inspecting tags, selecting branches, following associative links to discover cues it could not have anticipated. This is not "searching multiple times" but navigating a heterogeneous structure with typed actions at each step. The novelty of MRAgent lies in **reasoning-guided traversal over structured associative memory**.
>
> We implemented an agentic RAG baseline (same LLM, same max reasoning turns T=8 and per-turn retrieval budget K=10, iterative retrieval over flat vector store). On LoCoMo(Gemini) evaluated by LLM-Judge:
> |Method|Multi-hop|Temporal|Open Domain|Single hop|Overall|
> |--|--|--|--|--|--|
> |RAG (one-shot)|58.16|49.22|41.67|69.20|61.30|
> |Agentic RAG (Search-o1)|63.12|59.50|40.62|71.82|65.71|
> |MRAgent|75.17|80.37|68.75|90.48 |84.21|
>
> Multi-step flat retrieval falls far short of MRAgent.
>
> **W2: (Comparison with Graph memory)** We already discuss graph-based memory in Section 2 and include A-Mem as a graph baseline to represent a common class of approaches that rely on similarity-based seeding with predefined expansion.  We will expand the related work to cover H-MEM and SGMEM. The key differences:
> |Method|Node Types|Edge Types|Retrieval|ACTIVE|
> |--|--|--|--|--|
> |A-Mem|note|semantic link|similarity+N-hop|No|
> |SGMEM|sentence|similarity|similarity+N-hop|No|
> |H-MEM|multi-granularity|position index|similarity| No|
> |MRAgent|cue, content|tag-typed relation|**LLM-guided reasoning**|**Yes**|
>
> Prior graph methods use similarity seeding + predefined expansion — retrieval remains fixed. In MRAgent, the LLM actively determines traversal direction based on intermediate evidence. Tags provide a decision interface to preview, select, and revise exploration paths before accessing content. The difference is not the graph structure itself, but the combination of structured associative memory and reasoning-guided reconstruction. H-MEM and SGMEM do not provide public code. More importantly, we view these methods as belonging to the same class of graph-based retrieval systems, with A-Mem as its representative baseline. MemoryOS also adopts a hierarchical structure similar to H-MEM and is included in our baselines.
>
> **W3: (Token cost analysis)**  Thank you for raising this. Table 3 does include both construction and retrieval costs (as noted in the caption).  Methods such as A-Mem repeatedly summarize history and analyze complex dependencies during graph construction. In contrast, MRAgent adopts a lightweight construction phase and LLM actively explores the graph. This reduces preprocessing while enabling more efficient retrieval.
>
> We appreciate the reviewer's suggestion of relevant works, and we would like to emphasize that the novelty of our work is not the use of graph memory or multi-step retrieval, but a new abstraction of memory access: **retrieval as reasoning-guided traversal over an associative Cue–Tag–Content structure**, where intermediate evidence directly determines the next retrieval operator rather than only refining the query. This formulation is not captured by agentic RAG (which adapts queries over flat retrieval) nor by prior graph memory systems (which rely on predefined or similarity retrieval).

---

> > ### Author Rebuttal · Reviewer_oJai · 2026-04-04
> >
> > Thank you for the new results. I'll increase my score acordingly.

---

> > > ### Author Response · Authors · 2026-04-07
> > >
> > > We sincerely thank you for the positive feedback and for raising the score!  The insightful suggestions have greatly helped us improve the quality of our manuscript. We will incorporate all the suggestions into the revised manuscript.

---

### Official Review · Reviewer_AjqA · 2026-03-12

**Soundness:** 3
**Presentation:** 2
**Significance:** 3
**Originality:** 3
**Overall Recommendation:** 4
**Confidence:** 4

**Summary:**

This manuscript proposes MRAgent, a memory-augmented agent framework to assist LLMs in reasoning over long interaction histories. MRAgent employs multi-turn retrieval to iteratively extract the necessary memories from the cue-tag-semantic graph. Experimental results show that the proposed method achieves better performance than existing memory-augmented agent approaches while consuming fewer tokens.

**Compliance With Llm Reviewing Policy:**

Affirmed.

**Final Justification:**

The authors have adequately addressed my initial concerns during the rebuttal phase, particularly regarding the unclear notations and limited methodological descriptions.
The proposed method seems to be novel and scalable, and empirically outperforms the baselines. However, my assessment of the paper's clarity remains a minor concern. Since the original submission requires non-trivial revisions to incorporate the corrected notations and the expanded explanations (both on the method and related works) discussed during the rebuttal, my final recommendation is a Weak Accept. I strongly encourage the authors to rigorously integrate all these promised improvements.

**Key Questions For Authors:**

Q1. How is the cue-tag-semantic graph constructed?

Q2. Why is Theorem 1 non-trivial? Additional clarification would help readers understand its significance.

Q3. Please clarify the issues in Weaknesses.

**Limitations:**

The manuscript does not explicitly discuss limitations. A potential limitation is that the graph construction process does not appear to be automated, which may limit the scalability or applicability of the approach.

**Strengths And Weaknesses:**

### Strengths

#1. The proposed multi-turn memory reconstruction enables efficient extraction of relevant memories and appears to be a reasonable methodology.

#2. The experiments demonstrate that the method outperforms existing memory-augmented agent approaches in terms of both performance and scalability.

#3. The figures are well designed and help readers understand the overall framework.

### Weaknesses

#1. Lack of rigor in mathematical notation.
- Duplicate use of notation:
  - $\mathcal{C}$ is used as a cue set, but Eq. (9) also uses the notation $\mathcal{C}\_\text{LLM}$ to denote a function.
  - $\mathcal{R}$ is used as a triplet set, yet Eq. (11) introduces $\mathcal{R}\_\text{LLM}$ as a function.
  - In Eq. (7), $c$ appears both as a subscript of $Pi$ and $\phi$, and simultaneously as an element of $\mathcal{C}^{(t)}$ as well as an input to $\phi$. Similar reuse of notation also appears in Eq. (8).
- Undefined notation: $\mathcal{C}\_\text{LLM}$ and $\mathcal{R}\_\text{LLM}$ are not formally defined in the manuscript.

#2. The manuscript does not describe how the cue-tag-semantic graph is constructed. Since this graph is a key component of the proposed method, the lack of description on the cue-tag-semantic graph makes it difficult to understand and reproduce the approach. It is also unclear how the method can be applied in practice.

#3. Theorem 1 claims that multi-turn retrieval can capture more information than single-step retrieval, which appears self-evident. Clarification on why the theorem is non-trivial and additional justification for its theoretical contribution would be helpful.

#4. The related work section is very limited and consists of only a single paragraph. A broader discussion of relevant literature is necessary to properly position this work.

#5. The manscript does not state how episodic memory and semantic memory are used in Section 4. Also,  a more detailed explanation on how $\mathcal{H}^{(t+1)}$ is evaluated by $\mathcal{C}_\text{LLM}$ is needed.

#6. Minor: Line 249 states that mapping operator $\phi$ for Content$\to$ (Cue,Tag) is defined in Section 3, which could not be located.

---

> ### Author Rebuttal · Authors · 2026-03-30
>
> We thank the reviewer for the detailed feedback and for recognizing the effectiveness and scalability of our approach. We address each concern below.
>
> **W1/W6: (Mathematical Notation)** We sincerely apologize for the inconsistencies. We provide a complete resolution for each issue, and will incorporate all changes in the revised manuscript.
>
> - C overloaded (cue set vs. Eq.9 function): renamed to C for the cue set, $f_{select}$ for the action-selection function in Eq.(9).
> - R overloaded (relation set vs. Eq.11 function): renamed to R for relations, $f_{route}$ for the routing function in Eq.(11).
> - c in Eq.(7)–(8) used as both subscript and set element: introduced distinct notation c' for iteration variable vs. c for indexing.
> - C_LLM and R_LLM: now formally defined at first use in Section 4.2 as the action-selection and routing functions respectively.
> - Line 249: corrected to reference Eq.(8), Section 4.1.
>
> **W2/Q1/Limitations: (Graph Construction Description)** Due to page limits, the full pipeline of Graph Construction was in Appendix B.1 (Eq. 17–20) . We will move it into Section 3 in the revision. This graph is **fully automated via LLM extraction**(no manual annotation), using a fixed prompt template for cue/tag extraction, and scales linearly with the dialogue length.
>
> (1) Episodic construction: Dialogues are first processed with pronoun resolution and temporal normalization, then segmented into episodic units. For each episode $e_i$, an LLM extracts:
>
> * A tag $g_i$: a short phrase summarizing the core relation,
> * A cue set $C_i$ : fine-grained entities, contextual keywords.
>
> These form Cue–Tag–Episode triplets.
>
> (2) Semantic construction: A separate LLM call extracts stable knowledge as semantic triplets $(c_i^s, g_i^s, s_i)$, where $c_i^s $ is an entity-level cue, $g_i^s $ is an aspect-level tag, and $s_i $ is the semantic content.
>
> We will include this in the revised main text.
>
> **W3/Q2: (Theorem Contribution)** The theorem does **not** claim that "more retrieval steps are better than fewer." It proves that under the **same** budget T, active retrieval is strictly more powerful than passive retrieval in the memory-retrieval setting — to our knowledge, the first such formalization for LLM agents interacting with graph-structured memory.
>
> **The result is quantitative, not just qualitative:**  passive retrieval incurs irreducible error at least $ε_Y(1−T/2^d)$ on our constructed task family, while active retrieval achieves zero error with budget T = d+1. Matching this performance passively would require a budget on the order of 2^d. This exponential gap motivates MRAgent’s choice of an multi-turn reconstruction architecture over single-turn large-scale parallel retrieval(existing methods) and explains our experimental finding that deeper reconstruction yields monotonic gains while broader parallel retrieval saturates quickly(Figure 9) .
>
> **The theorem also guided concrete design choices:** (1) Internal nodes in the binary-tree construction encode navigational guidance rather than final answers. This motivates the Tag layer as lightweight navigational signals that allow the LLM to assess retrieval directions before committing to expensive episodic content.   (2) **LLM-guided adaptive retrieval, not merely multi-step retrieval.** The theorem shows that the gain comes from conditioning each retrieval decision on accumulated evidence, which motivates our three-stage loop (LLM Reasoning → Controlled Traversal → LLM Routing), where the LLM actively controls retrieval direction.
>
> **W4: (Related Work)** We agree that a more comprehensive related work discussion is necessary. In the revision, we will expand this section to include:
>
> 1. RAG and its extensions: GraphRAG, Agentic RAG.
> 2. Graph-based memory systems: AriGraph, Zep, LiCoMemory.
> 3. Hierarchical and persistent memory systems: MemoryOS, Mem0, SeCom.
>
> We will position MRAgent as active reconstruction over structured memory against (a) retrieval over flat memory indices (RAG), (b) fixed graph expansion from similarity-based seeds (A-Mem, Zep), and (c) hierarchical but still passive access (MemoryOS).
>
> **W5: (Explanation of Memory Use and Eq.9)** Both episodic memory and semantic memory  are accessed through the same unified traversal framework (Eq. 7–8). The LLM adaptively selects which memory layer to access via distinct traversal actions detailed in Table 4 (Appendix B.2). Table 6 (Appendix D.7) shows that different retrieval operators specialize across query types, confirming that the framework enables adaptive, query-dependent memory access.
>
> In practice, Eq.(9) is implemented as a prompt-based decision: the LLM is presented with the query $x $, the current context $\mathcal{H}^{(t+1)} $ and candidate nodes.  At each step, the LLM makes a discrete decision between continuing traversal (`Navigate`) and terminating with an answer (`Answer`), based on whether the accumulated context is sufficient in Algorithm 1.
>
> This will be clarified in Section 4.2.

---

> > ### Author Rebuttal · Reviewer_AjqA · 2026-04-02
> >
> > The authors have addressed my concerns, and I have raised my score.

---

> > > ### Author Response · Authors · 2026-04-03
> > >
> > > We sincerely thank the reviewer for the constructive feedback and for raising the score. Your insightful comments have significantly contributed to the improvement of our manuscript. We will rigorously integrate all promised improvements in the revision:
> > >
> > > 1. Notation: All notational corrections discussed during the rebuttal (symbol disambiguation, formal definitions​, reference fixes) will be applied throughout the manuscript.
> > >
> > > 2. Method description: The graph construction pipeline (Appendix B.1) and the reconstruction algorithm with pseudocode (Appendix B.2) are already fully detailed in the appendix. We will move the key content into the main text (Sections 3–4) to make the paper self-contained.
> > >
> > > 3. Related work: We have provided a detailed expanded discussion in our response to Reviewer oJai, covering RAG extensions, graph-based memory systems, and hierarchical memory approaches. This will be incorporated into the revised manuscript.
> > >
> > > We are committed to ensuring that the revised version reflects all improvements discussed during the rebuttal.

---

### Official Review · Reviewer_Gm4s · 2026-03-14

**Soundness:** 3
**Presentation:** 3
**Significance:** 2
**Originality:** 2
**Overall Recommendation:** 4
**Confidence:** 3

**Summary:**

The paper moves away from the standard "retrieve-then-reason" approach, where memory is treated like a static search engine. Instead, the authors argue that for LLM agents to handle long histories, they need to "reconstruct" memory step-by-step.

**Compliance With Llm Reviewing Policy:**

Affirmed.

**Final Justification:**

I keep my score. I still see some issues with the experiments.

**Key Questions For Authors:**

Please see above.

**Limitations:**

Please see above.

**Strengths And Weaknesses:**

Strengths And Weaknesses:
The Tagging layer is a great middle-ground: Most memory systems either give the model too much raw text (too noisy) or too little summary (missing details). Using Tags as a "table of contents" for the agent to browse before committing to reading the full content is a clever efficiency play.
Massive Token Savings: The efficiency gains are hard to ignore. Dropping token usage from 632k down to 118k while actually improving accuracy on benchmarks is a strong signal that this architecture is onto something.

    However, there are still several concerns left.
Core Ablation Weakness: Graph vs. Agent Loop: The experiments do not clearly show whether the performance gain comes from the Cue–Tag–Content graph structure or simply from the multi-step agentic retrieval loop. The paper only compares MRAgent (graph + multi-step retrieval) with standard single-step RAG systems, which is an unfair comparison.
Theoretical Contribution is Trivial: Theorem 4.1 claims that active retrieval is more expressive than passive retrieval, which is essentially obvious: a system that can adapt decisions sequentially will always have a larger hypothesis space than one making a single decision.
Unanalyzed Cascading Failures: MRAgent performs greedy traversal and pruning over the graph. If the LLM makes an incorrect decision early (e.g., selecting the wrong Tag and pruning the correct branch), the system may irreversibly deviate from the correct path.

---

> ### Author Rebuttal · Authors · 2026-03-30
>
> We thank the reviewer for recognizing the Tag layer as an effective and efficient mechanism and the significance of the token savings (632k→118k with improved accuracy), and for the constructive feedback that helped us strengthen the paper.
>
> **W1: (Graph vs. Agent Loop)**  We thank the reviewer for this important concern. We respectfully note that Fig. 5 already separates these effects: CE→CTE→CTC (green bars) shows the contribution of memory structure alone, while the improvement from CTC to MRAgent isolates the gain from reasoning-guided reconstruction.
>
> More importantly, we would like to clarify that MRAgent is **not** "graph + multi-step retrieval." Rather, the key contribution is a reasoning-guided reconstruction process over structured associative memory. The structured memory and the reasoning process are **tightly coupled**: without CTC, the agent cannot preview directions; without reasoning, the graph reduces to fixed expansion. The gain comes from their integration.
>
> We would also like to clarify that **multi-step reasoning in MRAgent is different from multi-step retrieval**. MRAgent is not repeated retrieval over a graph. It is adaptive reasoning over associative structure, where Tags expose retrieval directions before expensive content access.
>
> To directly test the reviewer’s hypothesis,  we implemented a multi-step RAG baseline using the same LLM with iterative retrieval over a flat vector store.
>
> | Method         | Multi-hop | Temporal | Open Domain | Single hop | Overall |
> | -------------- | --------- | -------- | ----------- | ---------- | ------- |
> | RAG (one-shot) | 58.16      | 49.22    | 41.67       | 69.20      | 61.30   |
> | Multi-step RAG | 63.12     | 59.50    | 40.62       | 71.82      | 65.71   |
> | MRAgent        | 75.17     | 80.37    | 68.75       | 90.48      | 84.21   |
>
> Multi-step retrieval over flat memory helps moderately but falls far short of MRAgent.
>
> **W2: (Theoretical Contribution)** We thank the reviewer for this comment. While the high-level intuition that “adaptive > non-adaptive” is natural, Theorem 4.1 formalizes this gap under the **same budget T** and establishes a strict separation in the memory-retrieval setting.
>
> **The result is quantitative, not just qualitative:**  passive retrieval incurs irreducible error at least $ε_Y(1−T/2^d)$ on our constructed task family, while active retrieval achieves zero error with budget T = d+1. Matching this performance passively would require a budget on the order of 2^d. This exponential gap directly motivates MRAgent’s choice of an iterative multi-turn reconstruction architecture over single-turn large-scale parallel retrieval(existing methods) and explains our experimental finding that deeper reconstruction yields monotonic gains while broader parallel retrieval saturates quickly(Figure 9) .
>
> **The theorem also guided concrete design choices:** (1) Internal nodes in the binary-tree construction encode navigational guidance rather than final answers. This motivates the Tag layer, where Tags serve as lightweight navigational signals that allow the LLM to assess retrieval directions before committing to expensive episodic content.   (2) **LLM-guided adaptive retrieval.** The theorem shows that the gain comes from conditioning each retrieval decision on accumulated evidence, which motivates our three-stage loop (LLM Reasoning → Controlled Traversal → LLM Routing), where the LLM actively controls retrieval direction.
>
> **W3: (Cascading Failure Analysis)** MRAgent mitigates cascading failure risk through several mechanisms: (1) multiple candidate nodes are maintained at each step, allowing exploration diversity rather than single-path selection; (2) traversal decisions are conditioned on accumulated evidence, enabling the model to revise or redirect its trajectory; (3) parallel and multi-path exploration is supported, where the agent can follow multiple branches simultaneously; (4) the CTC graph provides rich associative connectivity with alternative paths to the same evidence (as illustrated in Fig. 7).
>
> We conducted a conditional analysis on LoCoMo multi-hop questions, splitting queries by whether the agent retrieved valid evidence in the first round, and tracking per-group cumulative evidence recall:
>
> | Group        | Rd0   | Rd1   | Rd2   | Rd3   | Rd4   |
> | ------------ | ----- | ----- | ----- | ----- | ----- |
> | Round 0 hit  | 71.0% | 77.8% | 81.3% | 83.4% | 85.3% |
> | Round 0 miss | 0.0%  | 44.6% | 57.1% | 63.1% | 67.1% |
>
> Thus, early mistakes are not irreversible: even Round-0 misses recover to 67.1% recall after later rounds.

---

> > ### Author Rebuttal · Reviewer_Gm4s · 2026-04-03
> >
> > Thanks for the new results

---

> > > ### Author Response · Authors · 2026-04-07
> > >
> > > We thank the reviewer for the encouraging feedback and for acknowledging our additional experiments. We are glad the new results helped address the concerns. We look forward to any follow-up questions and are happy to provide further clarification.

---

### Decision · Program_Chairs · 2026-04-30

**Decision:**

Accept (regular)

**Comment:**

MRAgent reframes memory access as reasoning-guided reconstruction over a Cue–Tag–Content graph, where the LLM iteratively traverses structured associative memory rather than performing one-shot retrieval. Reviewers consistently praised the clarity of the abstraction, the substantial token-efficiency gains, and the strong empirical results on long-horizon memory benchmarks. The rebuttal meaningfully addressed concerns around fair comparison with agentic RAG, notation rigor, and robustness to upstream extraction errors via controlled perturbation experiments. All four reviewers converged on weak accept; I recommend acceptance.